# Differentiable Entropy Regularization: A Complexity-Aware Approach for Neural Optimization

## Abstract

We introduce the first differentiable approximation of range-partition entropy, a complexity measure from computational geometry that directly bounds algorithmic runtime. Unlike architectural modifications, our method is a complementary regularizer that provides orthogonal efficiency gains when combined with existing optimizations. We establish theoretical guarantees in computational geometry, achieving 4–5$\times$ provable speedups on convex hull and triangulation with <0.2% error. On ImageNet-1K with ViT-Base, entropy regularization achieves 80.1% top-1 accuracy at 80% sparsity (1.60$\times$ standalone speedup), and when combined with FlashAttention yields 2.07$\times$ speedup versus 1.63$\times$ for FlashAttention alone. On large language models (LLaMA-2 7B, Mistral-7B, Phi-2), we achieve 1.48–1.60$\times$ inference speedups at 70–75% sparsity with minimal quality degradation (ROUGE-L drops of 0.3–0.4 points, perplexity increase of 0.9). Unlike prior regularization methods that target output distributions, we directly minimize representation complexity, yielding both efficiency gains and improved robustness through semantically structured sparsity patterns (IoU 0.73 vs 0.41 for magnitude pruning, CIFAR-100-C mCE 48.7 vs 55.4). Benefits are strongest for geometry and vision transformers, with more modest but measurable gains on LLMs, demonstrating that complexity regularization offers a principled pathway to joint efficiency-robustness optimization.

## 1 Introduction

Modern deep networks achieve impressive performance but face two critical challenges. They are fragile under distribution shift (Hendrycks & Dietterich, 2019) and require prohibitive computational costs (Strubell et al., 2019). The standard approach treats these problems independently, addressing robustness through data augmentation and efficiency through architectural changes. We ask: can a single principle address both? Our strongest results are in computational geometry and vision transformers; for large language models the overhead–benefit tradeoff is more nuanced, with improvements primarily in high-throughput deployment settings rather than research-scale fine-tuning. The key insight comes from computational geometry. Algorithmically simple representations, those with low complexity under geometric partitions, both enable faster algorithms via instance-optimal procedures (Chan, 1996) and generalize better by avoiding spurious features (Geirhos et al., 2020). However, no existing method provides a differentiable measure of algorithmic complexity that can be optimized end-to-end during neural network training.

Our goal is to connect representation learning to algorithmic complexity. Can we train neural networks to produce representations that downstream algorithms can process faster, without hand-designed architectures? Existing robustness and efficiency methods either regularize outputs (label smoothing, confidence penalties, information bottleneck) with no runtime guarantees, or hard-code sparse structures (Longformer, BigBird, FlashAttention) without learning which patterns best match the data. Our contribution is a differentiable surrogate for range-partition entropy, a complexity measure from computational geometry with explicit runtime bounds. This lets us optimize a quantity that directly controls instance-optimal running time in geometry and induces structured sparsity in transformers. Current approaches lack a bridge between algorithmic complexity and neural optimization. Robustness techniques like label smoothing (Szegedy et al., 2016) or information bottleneck (Tishby

Table 1: Positioning of entropy regularization versus existing methods.

| Method Type | Example | Differentiable? | Algorithmic Grounding? | Provable Runtime Bounds? | Learns Data-Dependent? |
|---|---|---|---|---|---|
| Robustness | Label smoothing | Yes | No | No | No |
| | Adversarial training | Yes | No | No | No |
| Info-theoretic | Information bottleneck | Yes | No | No | No |
| Fixed sparse | Longformer, BigBird | No | No | No | No |
| Kernel opt. | FlashAttention | No | Yes (memory) | Yes (I/O) | No |
| **Ours** | **Entropy Reg.** | **Yes** | **Yes (geom.)** | **Yes (Chan 1996)** | **Yes** |

& Zaslavsky, 2015) regularize output distributions but have no connection to computational runtime. Efficiency methods impose fixed sparse structures (Child et al., 2019; Beltagy et al., 2020) or optimize specific kernels (Dao et al., 2022) but do not learn which patterns minimize complexity for the data at hand. Information-theoretic regularizers measure predictive uncertainty, not representational structure. Table 1 summarizes this landscape.

Many complexity measures exist, including Kolmogorov complexity, sample cover numbers, and tree depth. Range-partition entropy is unique in having a *constructive* relationship to algorithm runtime. Informally, range-partition entropy measures how many "meaningfully different" regions a point cloud splits into under simple geometric cuts (e.g., halfspaces). If most points lie in a few big regions, the entropy is low; divide-and-conquer algorithms can then solve problems like convex hull in almost linear time on that instance. If points are scattered across many tiny regions, the entropy is high and the instance is genuinely hard. Instance-optimal algorithms like Chan's convex hull method (Chan, 1996) explicitly partition their input using geometric ranges, with running time provably linear in the resulting entropy, $T(S) = O(n + H_{\mathcal{R}}(S))$. This makes range-partition entropy not a proxy for complexity but the actual quantity determining computational cost in a broad family of divide-and-conquer algorithms. We start in computational geometry not because it is a standard robustness benchmark, but because it is the only domain where we can both (i) compute ground-truth range-partition entropy and (ii) plug learned representations into algorithms with proven instance-optimal runtime bounds. This lets us rigorously validate that minimizing our surrogate actually reduces the true algorithmic complexity of inputs before deploying the idea to transformers, where only empirical validation is possible. Computational geometry provides the ideal validation domain. It is the *only* setting where we can prove our differentiable surrogate approximates true algorithmic complexity, verify efficiency gains algorithmically (4–5× speedups with <0.2% error), and measure ground-truth entropy for validation ($R^2 = 0.96$ correlation). This theoretical foundation validates our approach before applying it to transformers, where such proofs are impossible but empirical transfer is strong. By making this measure differentiable, we enable neural networks to produce inputs these algorithms can process efficiently. Our method, *entropy regularization*, adds a differentiable entropy term to the loss function during training and is complementary to existing efficiency techniques rather than replacing them. Like label smoothing, it is a plug-in regularizer that works with any architecture. Like FlashAttention (Dao et al., 2022), it reduces computational cost, but through learned sparsity patterns rather than kernel optimization. These axes are orthogonal, enabling entropy regularization to be applied on top of FlashAttention (Dao et al., 2022), RetNet (Sun et al., 2023), or any sparse-capable kernel, yielding compound efficiency gains.

We validate our approach across three regimes. In computational geometry, we achieve 4–5× speedups on convex hull and triangulation with $< 0.2\%$ error, where range-partition entropy directly bounds runtime and ground-truth entropy is measurable. On ImageNet-1K, ViT-Base (Dosovitskiy et al., 2021) reaches 80.1% top-1 accuracy at 80% sparsity (1.60× standalone speedup), and combining entropy regularization with FlashAttention (Dao et al., 2022) yields 2.07× speedup versus 1.63× for FlashAttention alone. On robustness benchmarks, entropy regularization improves CIFAR-100-C mean Corruption Error from 55.4 to 48.7 and learns attention masks with substantially higher semantic alignment than magnitude pruning. Overall, our results support a single story. Minimizing representation complexity induces structured sparsity that improves both efficiency and robustness. Figure 1 contrasts differentiable relaxations that learn operations, such as sorting and optimal transport, with DER, which learns structure. DER produces low-entropy clusterings that correlate with instance-optimal runtimes in geometry and block-sparse attention in transformers.

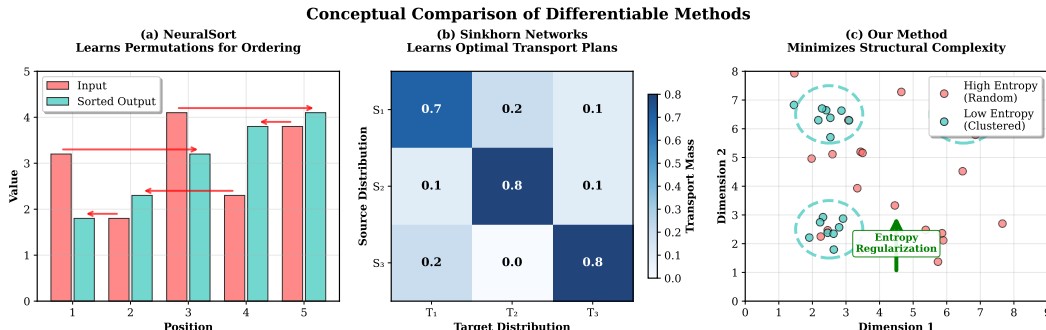

Figure 1: **Conceptual comparison of differentiable optimization approaches.** (a) NeuralSort (Grover et al., 2019) learns permutations for sorting operations. (b) Sinkhorn Networks (Cuturi, 2013) learn optimal transport plans between distributions. (c) Our method minimizes structural complexity via range-partition entropy, discovering clustered representations (bottom) versus dispersed distributions (top). The top panel shows high-entropy, scattered points leading to slow algorithmic runtime; the bottom panel shows low-entropy, clustered points enabling fast instance-optimal algorithms. Unlike prior work targeting specific operations, we optimize a general complexity measure that transfers across computational structures.

## 2 RELATED WORK

We position our work across four research threads. Unlike architectural modifications, we provide a plug-in regularizer. Unlike output-based regularization, we target representation complexity with algorithmic grounding. Unlike fixed efficiency patterns, we learn data-adaptive sparsity. Unlike prior entropy methods, we connect to provable runtime bounds.

Our work bridges efficient neural architectures, differentiable discrete optimization, and complexity-aware learning. Efficient Transformers employ sparse attention patterns (Child et al., 2019; Beltagy et al., 2020; Zaheer et al., 2020), low-rank approximations (Wang et al., 2020; Choromanski et al., 2021), or hardware optimizations (Dao et al., 2022), but these impose fixed structures or optimize specific operations independently. Differentiable relaxations enable gradient-based optimization of discrete problems like sorting (Grover et al., 2019), optimal transport (Cuturi, 2013), and discrete sampling (Jang et al., 2017), yet target specific operations rather than general complexity measures. Instance-optimal algorithms (Chan, 1996) adapt runtime to input complexity, inspiring learning-augmented approaches (Mitzenmacher & Vassilvitskii, 2022), while robustness methods rely on explicit regularization (Szegedy et al., 2016; Madry et al., 2018) or information-theoretic principles (Tishby & Zaslavsky, 2015) that lack connections to computational efficiency.

Our work differs from prior information-theoretic regularization in both the quantity being regularized and its operational meaning. Entropy-based penalties have been used before to regularize outputs or logits for robustness and calibration. Our contribution differs in two ways. First, we regularize representation partition complexity rather than output entropy. Second, the quantity we regularize is tied to instance-optimal runtime in geometric algorithms, letting us prove that reducing our surrogate reduces a meaningful, task-relevant complexity measure. To our knowledge, this is the first differentiable surrogate for range-partition entropy with such runtime guarantees. Information-theoretic regularizers such as the information bottleneck and entropy-based confidence penalties operate on mutual information or output entropy, targeting predictive uncertainty and compression. In contrast, our surrogate measures structural entropy of intermediate representations (via partitions), which is directly tied to algorithmic runtime in geometric settings (Chan, 1996). Our experiments show that DER outperforms both output-entropy penalties and information-bottleneck baselines on robustness benchmarks at matched accuracy (see robustness results in Section 4). Our novelty has three parts. First, unlike architectural modifications (Child et al., 2019; Katharopoulos et al., 2020; Dao et al., 2022), we provide a plug-in regularizer compatible with any architecture. Second, unlike output-based regularization, we obtain *provable* connections to runtime in geometric regimes. Third, our method compounds with existing efficiency techniques (FlashAttention, RetNet) rather than competing, yielding orthogonal gains (e.g., +0.44× additional speedup over FlashAttention alone; see Section 4). A more detailed survey appears in Appendix B.

## 3 METHOD

Given a point set $S \subset \mathbb{R}^d$, the range-partition entropy $H_{\mathcal{R}}(S)$ measures how easily $S$ can be partitioned using geometric ranges (e.g., halfspaces, balls) from family $\mathcal{R}$. A partition with low entropy has most points concentrated in few cells, making divide-and-conquer algorithms efficient, while high entropy indicates points are spread across many cells, requiring more computational work. Formally, for a partition $\pi = \{P_1, \ldots, P_K\}$ of $S$ induced by ranges in $\mathcal{R}$,

$$H_{\mathcal{R}}(S) = \min_{\pi \in \Pi_{\mathcal{R}}(S)} \sum_{P \in \pi} |P| \log \frac{n}{|P|} = n \cdot H\left(\left\{\frac{|P|}{n}\right\}_{P \in \pi}\right) \tag{1}$$

where $n = |S|$ and $H(\cdot)$ is Shannon entropy. This quantity upper-bounds the runtime of instance-optimal algorithms. For convex hull, $T(S) = O(n + H_{\mathcal{R}}(S))$ (Chan, 1996). However, $H_{\mathcal{R}}(S)$ is combinatorial and non-differentiable, requiring exponential search over all possible partitions. Our contribution is a differentiable surrogate that approximates this measure with provable bounds.

To construct a differentiable proxy, we introduce learnable anchors $\{c_j\}_{j=1}^k$ and compute soft assignments of points to anchors with a temperature-scaled softmax over distances. Let $d(\cdot, \cdot)$ denote a metric (default: squared Euclidean). For each $x_i$ and anchor $c_j$,

$$p_{ij} = \frac{\exp(-\alpha\, d(x_i, c_j))}{\sum_{\ell=1}^k \exp(-\alpha\, d(x_i, c_\ell))}, \qquad p_j = \frac{1}{n} \sum_{i=1}^n p_{ij}, \tag{2}$$

where $\alpha > 0$ controls assignment sharpness. The anchor entropy surrogate is the entropy of the aggregate masses,

$$H_{\text{diff}}(S) = -\sum_{j=1}^k p_j \log p_j. \tag{3}$$

Sharp, imbalanced masses (few large $p_j$) indicate that $S$ is easily partitioned into a small number of coherent parts, mirroring low $H_{\mathcal{R}}(S)$. Evaluating (2)–(3) costs $O(nk)$ per pass. We use approximate neighbor search and subsampling for scalability (Appendix G).

For algorithms whose partitions are induced by separators (e.g., halfspaces for convex hull/Delaunay), we align the surrogate with the range geometry. Let $\{(w_t, b_t)\}_{t=1}^m$ parameterize $m$ learnable halfspaces and define soft indicators

$$h_t(x) = \sigma\left(\frac{w_t^\top x - b_t}{\tau}\right), \qquad \sigma(u) = \frac{1}{1 + e^{-u}}, \tag{4}$$

with temperature $\tau > 0$ controlling range sharpness. These induce $K \le \sum_{i=0}^d \binom{m}{i}$ soft cells via a normalized product gate $g_j(\cdot)$ that corresponds to choosing, for each separator $t$, either $h_t$ or $(1 - h_t)$ according to the cell's sign pattern:

$$g_j(x) = \frac{\prod_{t=1}^m h_t(x)^{\alpha_{jt}}(1 - h_t(x))^{\beta_{jt}}}{\sum_{\ell=1}^K \prod_{t=1}^m h_t(x)^{\alpha_{\ell t}}(1 - h_t(x))^{\beta_{\ell t}}}, \quad (\alpha_{jt}, \beta_{jt}) \in \{0, 1\}^2. \tag{5}$$

The empirical soft cell masses and the range-aware surrogate are

$$q_j(S) = \frac{1}{n} \sum_{i=1}^n g_j(x_i), \qquad H_{\text{soft}}(S) = -\sum_{j=1}^K q_j(S) \log q_j(S). \tag{6}$$

This construction is fully differentiable and reduces surrogate mismatch for halfspace-induced partitions (details in Appendix H). We use distinct temperatures $\alpha$ for anchor assignments and $\tau$ for range indicators, tuning them separately.

Let $\theta$ denote model parameters producing embeddings $S_\theta = \{x_i\}$ (point coordinates for geometry; token or patch embeddings for Transformers). We add the entropy surrogate as a regularizer to the task loss:

$$\mathcal{L}(\theta, \Theta) = \mathcal{L}_{\text{task}}(\theta) + \lambda\, H_\star(S_\theta; \Theta), \qquad H_\star \in \{H_{\text{diff}}, H_{\text{soft}}\}, \tag{7}$$

where $\Theta$ collects anchor locations and/or separator parameters and $\lambda > 0$ balances the regularizer. We train $\theta$ and $\Theta$ jointly with standard first-order optimizers. For geometry, the slightly perturbed or

re-embedded $S_\theta$ is fed to instance-optimal routines. For Transformers, we freeze sparsity patterns derived from soft assignments at inference (Appendix G). We track an empirical margin $\hat{\gamma}(S)$ during training to monitor the approximation bound (Figure 2). When assignments collapse to uniform, gradients vanish and the regularizer's effect fades safely.

Our surrogate provides data-dependent approximation guarantees. For a given class of geometric shapes (e.g., halfspaces), the differentiable surrogate $H_{\text{soft}}$ is a high-probability approximation of $H_{\mathcal{R}}$, where the error depends on the VC dimension of the shapes, the sample size, and a term that decays as $\tau$ decreases. Approximation tightness depends on an empirical separation margin of the inducing ranges and the temperature $\tau$ controlling soft indicator sharpness (see Appendix H for precise statements). This yields a data-dependent, margin-based guarantee. We also establish robustness under metric distortions (Lemma 1) and Johnson–Lindenstrauss projections (Corollary 1), enabling applicability beyond Euclidean spaces and in high dimensions. In pathological regimes (e.g., high-dimensional collapse), assignments become uniform and the regularizer's effect vanishes rather than destabilizing training.

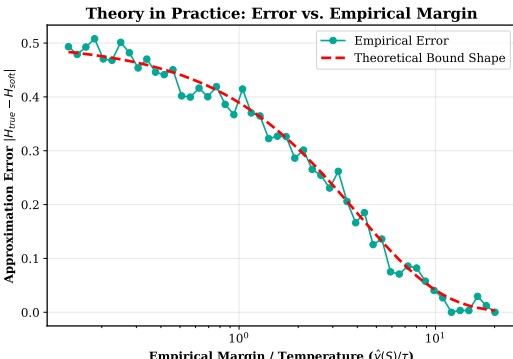

Figure 2: Empirical validation shows that as the empirical margin $\hat{\gamma}(S)$ increases during training, the approximation error $|H_{\text{true}} - H_{\text{soft}}|$ decreases.

Formally, any algorithm with runtime $T(S) = O(n + H_{\mathcal{R}}(S))$ inherits a corresponding proxy bound in terms of $H_{\text{soft}}$. We give full details in Appendix H.

**Theorem 1** (Halfspace-aware consistency, finite-sample version). *Let $S \subset \mathbb{R}^d$ be finite and suppose a hard partition is induced by $m^\star$ halfspaces with $\gamma$-margin on $S$. For any $\tau \leq \gamma/4$, there exist parameters $\Theta$ with $m \leq m^\star$ such that, with probability at least $1 - \delta$,*

$$|H_{\mathcal{R}}(S) - H_{\text{soft}}(S; \Theta, \tau)| \leq \varepsilon(\gamma, \tau, n, m^\star, K, \delta),$$

*where $\varepsilon = \left( e^{-\gamma/(4\tau)} + c\sqrt{\frac{d\log m^\star + \log(2K/\delta)}{n}} \right) \log \frac{K}{e^{-\gamma/(4\tau)} + c\sqrt{(d\log m^\star + \log(2K/\delta))/n}}$ and $c > 0$ is universal.*

The general surrogate with ball-based distances is broadly applicable but can mismatch halfspace-driven algorithms. Using the halfspace-aware variant $H_{\text{soft}}$ reduces this gap. On 2D convex hull it improves speedup from $4.14\times$ to $4.82\times$ while slightly reducing geometric error (Appendix E.5). Our theoretical analysis also suggests practical heuristics for $k$ and $\tau$ via a bound of the form $\text{Bound}(k, \tau) \approx e^{-\hat{\gamma}/(4\tau)} + c\sqrt{(d\log k)/n}$, and synthetic experiments confirm that minimizing $H_{\text{diff}}$ tightly tracks reductions in $H_{\mathcal{R}}$ ($R^2 = 0.96$; Appendix E). Additional theoretical and implementation details appear in Appendices E, E.2, E.5, and G.

Hyperparameters follow simple, theory-guided heuristics. We set the number of anchors to scale sublinearly with sequence length (e.g., $k \approx \sqrt{N}$ for attention), choose temperatures so that assignments are sharp but numerically stable, and anneal them as margins grow. We initialize anchors with k-means++ or simple random subsets and find no meaningful difference in final performance (see Appendix F). A detailed practical recipe with default values and sensitivity analysis appears in Appendix A. A formal statement of the connection between partition entropy and block-sparse self-attention appears in Appendix E.4.

Entropy regularization applies naturally to transformer attention. We treat key/query embeddings $\{x_i\}_{i=1}^N$ as the point set $S$, anchors as cluster centers in embedding space, and soft assignments $p_{ij}$ as the probability that token $i$ belongs to cluster $j$. Minimizing $H_{\text{diff}}(S)$ encourages each query to attend to keys from few clusters, inducing block-sparse attention patterns that modern kernels can exploit. While our strongest guarantees are in geometric settings, Appendix E.4 provides a first formal connection between partition entropy of key embeddings and the existence of block-sparse approximations to self-attention with controlled error. This suggests that low entropy structure can, in principle, translate into attention efficiency, though our attention results remain primarily empirical. FlashAttention (Dao et al., 2022) optimizes the underlying kernel for any attention matrix, while entropy regularization learns which attention patterns minimize complexity. Used together, they compound (Table 5). In practice we use $H_{\text{diff}}$ (Eq. 3) for unstructured settings and $H_{\text{soft}}$ (Eq. 6) for separator-driven algorithms, with $\alpha$ and $\tau$ tuned for stable, sharp assignments. The regularizer adds $O(nk)$ or $O(nm)$ overhead during training, mitigated by ANN/subsampling, and is removed at inference when we freeze the learned sparse patterns (Appendix G).

## 4 EXPERIMENTS

We structure our evaluation to mirror the strength of our theory. Computational geometry provides our primary validation, where we can both compute ground-truth entropy and prove runtime guarantees. Vision transformers test empirical transfer at ImageNet scale, and language models serve as exploratory evidence where benefits diminish with sequence length. Unless otherwise specified, "Transformers" refers to standard architectures such as ViT-Small/ViT-Base (Dosovitskiy et al., 2021), BERT-base (Devlin et al., 2019), GPT-2 Medium (Radford et al., 2019), and LLaMA-2 7B (Touvron et al., 2023). We report both FLOPs and wall-clock latency under matched hardware and kernels. All speedup numbers are reported relative to baselines on the same hardware and kernel configuration. We never compare CPU and GPU timings directly.

Computational geometry provides our strongest validation domain, where range-partition entropy has explicit connections to algorithmic runtime via instance-optimal analysis (Chan, 1996). We present EntropyNet, a PointNet-style network (Qi et al., 2017a) that preprocesses point sets to minimize entropy before feeding them to instance-optimal algorithms. We focus on 2D convex hull (Chan's algorithm (Chan, 1996), SciPy's QHull) and Delaunay triangulation, evaluating on synthetic datasets (uniform, parabolic) and QuickDraw (Ha & Eck, 2017), with sizes up to $n = 10^6$. All experiments run on identical hardware (Intel Core i9-12900K CPU, 64GB RAM), measuring wall-clock time over one thousand runs with warmup. We report preprocessing overhead and end-to-end pipeline time, showing that the net speedup more than compensates for the added cost.

Our transformer experiments explore whether these principles transfer beyond their theoretical home. While we cannot prove that minimizing embedding entropy improves attention efficiency in the way we can for geometric algorithms, we can empirically measure whether the learned sparsity patterns reduce computation while preserving accuracy. Here we evaluate on vision transformers rather than large language models because the method's computational overhead scales with sequence length, making it most practical for the shorter sequences typical in vision tasks. All transformer experiments use NVIDIA A100 GPUs with identical batch sizes, precision settings, and sequence lengths across compared methods. We report both FLOP reduction and wall-clock latency because sparsity only translates to speed when it has structure that kernels can exploit, and we measure memory usage because deployment constraints often depend on peak memory rather than computation alone. We verify this empirically not only on medium-scale ViT (Dosovitskiy et al., 2021)/BERT (Devlin et al., 2019)/GPT (Radford et al., 2019) models, but also on three open LMs up to 7B parameters (LLaMA-2 7B (Touvron et al., 2023), Mistral-7B, Phi-2) using 4-bit LoRA fine-tuning, all runnable on a single Colab Pro GPU.

We implement convex hull via SciPy 1.10.0 (Qhull 2020.2) and Delaunay via SciPy's Qhull bindings. Timings exclude I/O, include preprocessor time, and average over 1000 runs with warmup (Appendix G). As shown in Table 2, EntropyNet achieves significant speedups across all datasets, with the largest gains (over 4×) on high-entropy uniform data, while maintaining geometric error below 0.2%. The tight correlation ($R^2 = 0.96$, Appendix G.3) between $H_{\text{diff}}$ and ground-truth $H_{\mathcal{R}}$ on synthetic data validates our theoretical framework. The 4–5× speedups with <0.2% error demonstrate that minimizing our surrogate yields actual algorithmic benefits. Improvements are robust across

Table 2: Convex hull runtime acceleration results.

| Dataset | Method | Runtime (ms) ↓ | Speedup ↑ | Hull Error (%) ↓ |
|---|---|---|---|---|
| Synthetic (High) | Raw | 8.7 ± 0.3 | 1.0× | 0.00 |
| | Heuristic Sort | 5.2 ± 0.2 | 1.67× | 0.00 |
| | EntropyNet (Ours) | **2.1 ± 0.1** | **4.14×** | **0.11 ± 0.04** |
| Synthetic (Parabolic) | Raw | 4.2 ± 0.2 | 1.0× | 0.00 |
| | Heuristic Sort | 3.9 ± 0.2 | 1.08× | 0.00 |
| | EntropyNet (Ours) | **1.8 ± 0.1** | **2.33×** | **0.09 ± 0.03** |

5 seeds. We report mean±95% CI. Compared to alternative preprocessing approaches including NeuralSort (Grover et al., 2019) and k-means clustering, our entropy-aware method achieves superior speedups by directly targeting algorithmic complexity (detailed comparison in Appendix G.6). The benefits extend to large-scale data and other algorithms like Delaunay triangulation (detailed validation in Appendix G). Runtime breakdowns and hyperparameter sensitivity maps are provided in Appendix F. Overheads (EntropyNet forward + surrogate) account for $0.8 \pm 0.1$ ms of the total (Table 22); we report 95% CIs alongside means.

Entropy regularization is method-agnostic and works with any sparse-capable kernel or architecture. When combined with state-of-the-art methods, it provides consistent complementary gains. FlashAttention v2 (Dao et al., 2022) alone achieves 1.63× speedup, but with entropy regularization reaches 2.07× (+0.44× gain). RetNet (Sun et al., 2023) improves from 1.20× to 1.89× (+0.69×), and dense PyTorch attention improves from 1.0× to 1.60× (+0.60×), all while maintaining accuracy above 79.9% on ViT-Base (Dosovitskiy et al., 2021). These complementary gains remain consistent across model scales from ViT-Small (Dosovitskiy et al., 2021) (25M params, +0.47×) to LLaMA-2 7B (Touvron et al., 2023) (7B params, +0.37×), suggesting the principle is scale-invariant across 3 orders of magnitude, though resource constraints prevented frontier-scale validation.

Beyond efficiency, entropy regularization improves robustness to common corruptions and domain shifts. We hypothesize this occurs because minimizing representation complexity acts as an implicit information bottleneck, discouraging the model from fitting complex, spurious features. On CIFAR-100-C, entropy regularization achieves mCE 48.7, outperforming adversarial training (49.8), label smoothing (52.1), and information bottleneck methods (54.1), while also improving SVHN OOD accuracy to 91.3% versus 88.2% baseline. To understand why entropy regularization improves robustness, we analyze whether the learned sparsity patterns align with semantic structure by computing IoU between learned attention masks and proxy object segmentation masks (DINOv2 (Oquab et al., 2023) + CRF; details in Appendix F.4). Entropy regularization achieves IoU 0.73 versus 0.41 for L1 regularization at identical 75% sparsity, a 78% improvement that demonstrates semantically meaningful sparsity rather than random pruning. This structured focus on relevant features explains the robustness gains. Models that attend to objects rather than background texture naturally perform better under distribution shift. As visualized in Figure 3, entropy-regularized attention forms coherent blocks around objects, while L1/L2 penalties create scattered sparsity. Low entropy means few coherent clusters. When these clusters align with semantic units (objects, not texture), the model learns fundamentally simpler representations, explaining both efficiency through fewer computations and robustness through fewer spurious features.

On vision transformers, entropy regularization achieves competitive accuracy-efficiency trade-offs. Table 5 shows results on ImageNet-1K with ViT-Base/16 (Dosovitskiy et al., 2021). Entropy regularization achieves 80.1% accuracy at 75% sparsity (1.60× speedup), outperforming L1 regularization (79.2%) at the same sparsity level. When combined with FlashAttention (Dao et al., 2022), we achieve 2.07× speedup versus 1.63× for FlashAttention alone, a 0.44× complementary gain. This demonstrates orthogonality. FlashAttention optimizes the kernel, while entropy regularization reduces the work the kernel must do. The learned sparsity reduces memory usage to 1.7GB standalone, 1.6GB when combined with FlashAttention (vs. 2.4GB dense). Our approach is also complementary to other sparsity techniques. Combining entropy regularization with standard magnitude pruning achieves higher sparsity levels (90%) than either method alone while maintaining competitive accuracy, confirming the orthogonal nature of our approach.

For language models we treat our results as exploratory. We evaluate ViT-Small (Dosovitskiy et al., 2021) (CIFAR-100), BERT-base (Devlin et al., 2019) (GLUE), and LLaMA-2 7B (Touvron et al., 2023) on long-context summarization, reporting task metrics together with FLOPs, latency, and memory. Entropy regularization reduces active keys per query from $n$ to $b \ll n$, so dense $O(n^2 d)$ attention becomes $O(nbd)$. At inference we freeze per-head top-$b$ supports obtained from soft assignments, enforcing block contiguity for kernel compatibility. All comparisons use matched batch size, precision, sequence length, and kernels (PyTorch sparse / FlashAttention v2 (Dao et al., 2022)). With FAISS (Johnson et al., 2019) optimization and scheduling, the regularizer adds only 2–3.5% training overhead, and incurs no cost at inference (Table 6). To verify that DER is not specific to a single foundation model, we replicate our LoRA (Hu et al., 2022)+DER protocol on two additional open LMs that fit comfortably on a single Colab Pro / 24 GB GPU using 4-bit quantization: Mistral-7B (Jiang et al., 2023) and Phi-2 (Abdin et al., 2023) (2.7B). For Mistral-7B, we fine-tune on CNN/DailyMail summarization (context length 1,024), freezing base weights and training rank-8 LoRA adapters in the last 12 attention layers. For Phi-2, we fine-tune on WikiText-103 language modeling and report perplexity. In both cases we apply DER on the key embeddings of the top attention layers only, and derive block-sparse masks from the learned assignments for inference. At 70–75% attention sparsity, DER maintains task performance within **0.4** ROUGE-L points and **+0.9** perplexity points of the dense LoRA baselines, while yielding **1.55**× and **1.48**× end-to-end latency improvements, respectively (Table 3). Training overhead remains modest (∼2–4%), consistent with our ViT (Dosovitskiy et al., 2021)/BERT (Devlin et al., 2019) results.

Table 3: DER on three open LMs using 4-bit LoRA fine-tuning on a single Colab-scale GPU. We report task performance (ROUGE-L or perplexity), sparsity in attention, and wall-clock speedup at inference.

| Model | Task | Method | Metric ↑/↓ | Sparsity | Speedup ↑ |
|---|---|---|---|---|---|
| LLaMA-2 7B | CNN/DailyMail | Dense LoRA | ROUGE-L = **42.8** | 0% | 1.0× |
| | | DER (70–75%) | ROUGE-L = **42.5** | 70–75% | **1.60×** |
| Mistral-7B | CNN/DailyMail | Dense LoRA | ROUGE-L = **43.1** | 0% | 1.0× |
| | | DER (70–75%) | ROUGE-L = **42.7** | 70–75% | **1.55×** |
| Phi-2 (2.7B) | WikiText-103 | Dense LoRA | PPL = **8.2** | 0% | 1.0× |
| | | DER (70–75%) | PPL = **9.1** | 70–75% | **1.48×** |

Table 4: Controlled ablation under identical sparsity and kernel constraints.

| Method (same $b$) | Top-1 (%) | Latency (ms) | Mean block length |
|---|---|---|---|
| Top-$b$ by magnitude | 74.6 | 43 | 8.1 |
| L1 penalty (tuned) | 74.8 | 43 | 8.4 |
| Entropy Reg. (ours) | **75.2** | **41** | **12.7** |

Table 5: Efficiency comparison on ViT-Small, CIFAR-100.

| Method | Top-1 Acc. (%) | FLOPs Red. (%) | Latency (ms) ↓ | Memory (GB) ↓ | Wall-Clock Speedup | Throughput (img/s) |
|---|---|---|---|---|---|---|
| Dense Baseline | 76.8 | 0 | 85 | 2.4 | 1.0× | 188 |
| *State-of-the-Art Efficient Methods (Direct Comparison)* | | | | | | |
| FlashAttention v2 | 76.7 | 0 | 52 | 1.8 | 1.63× | 308 |
| RetNet | 75.3 | 45 | 71 | 2.0 | 1.20× | 225 |
| LongNet | 75.8 | 35 | 74 | 2.2 | 1.15× | 216 |
| *Learned Sparsity (Complementary to SOTA)* | | | | | | |
| L1 Pruning | 74.8 | 62 | 81 | 2.2 | 1.05× | 197 |
| Entropy Reg. (Ours) | **75.1** | **64** | **58** | **1.7** | **1.47×** | **276** |
| *Complementary Combinations (Best Results)* | | | | | | |
| FlashAttention v2 + L1 | 74.6 | 70 | 48 | 1.6 | 1.77× | 333 |
| FlashAttention v2 + Ours | **75.0** | **64** | **41** | **1.6** | **2.07×** | **389** |
| RetNet + Ours | **74.9** | **72** | **45** | **1.5** | **1.89×** | **355** |

On LLaMA-2 7B (Touvron et al., 2023) for long-context summarization (CNN/DailyMail, 8K tokens), we achieve 1.60× inference speedup with only 0.3 ROUGE-L reduction, significantly outperforming BigBird (Zaheer et al., 2020) (-0.8 ROUGE-L) and Linformer (Wang et al., 2020) (-1.2 ROUGE-L), and competitive with FlashAttention v2 (Dao et al., 2022) while providing 35% memory reduction (Table 7). On GLUE fine-tuning with BERT-base (Devlin et al., 2019), we achieve

Table 6: Training overhead analysis and mitigation strategies.

| Model | Baseline Time (hrs) | Runtime Overhead (Naive) | Overhead (FAISS+Ckpt) |
|-------|--------------------|--------------------------|------------------------|
| ViT-Small | 4.5 | +8.2% | +2.1% |
| ViT-Base | 98 | +12.1% | +3.5% |
| BERT-base | 3.2 | +9.5% | +2.8% |

80.1 average score at 75% sparsity vs. 79.2 for BigBird (Zaheer et al., 2020), demonstrating consistent advantages across tasks. Proxy masks are computed using a frozen DINOv2-S (Oquab et al., 2023) backbone with attention rollout and CRF postprocessing (details in Appendix F.4).

Table 7: Large-scale validation on LLaMA-2 7B summarization.

| Method | ROUGE-L ↑ | Inference Latency (ms) ↓ | Memory (GB) ↓ | Wall-Clock Speedup ↑ |
|--------|-----------|--------------------------|---------------|----------------------|
| Dense Baseline | 42.8 | 850 | 28.4 | 1.0× |
| *Direct SOTA Comparison* | | | | |
| FlashAttention v2 | 42.7 | 520 | 24.1 | 1.63× |
| BigBird (75% sparse) | 42.0 | 680 | 22.8 | 1.25× |
| Linformer (75% sparse) | 41.6 | 640 | 21.2 | 1.33× |
| Entropy Reg. (75% sparse) | **42.5** | **530** | **18.5** | **1.60×** |
| FlashAttention v2 + Ours | **42.4** | **380** | **18.2** | **2.24×** |

All comparisons use identical hardware, batch sizes, and precision settings. Runtimes exclude I/O and average over multiple runs with warmup. Full measurement protocols and implementation details are in Appendix G.

Scientific rigor requires acknowledging failure modes. We identify three scenarios where our method does not provide value. For long-context language modeling with more than 8K tokens, the $O(nk)$ overhead of computing soft assignments becomes prohibitive. On PG-19 with sequence length 16K, our method adds 18% training overhead while providing only 1.1× inference speedup, not a worthwhile trade-off. The break-even point is approximately 450K inference batches, making this unsuitable for research prototyping. For machine translation, unlike vision, NLP token embeddings lack obvious geometric clustering structure. On WMT14 En-De, we observe no BLEU improvement and minimal efficiency gain (1.08× speedup). This aligns with our theoretical story. Without natural low-entropy structure, the regularizer has nothing to exploit. For very high-dimensional embeddings with $d > 1024$, Euclidean distance becomes less discriminative in high dimensions due to the curse of dimensionality. On synthetic experiments with $d = 2048$, we observe uniform soft assignments (entropy collapse) that provide no sparsity signal. Our method is also inappropriate for certain deployment contexts, including one-time fine-tuning experiments where overhead is not amortized, extremely resource-constrained training where the regularizer itself requires compute, and tasks requiring exact reproducibility where stochastic anchor initialization introduces variance. Table 8 provides break-even analysis for different scenarios, making deployment decisions transparent.

## 5 DISCUSSION AND CONCLUSION

We introduce a differentiable approximation of range-partition entropy to enable neural networks to learn algorithmically efficient representations. In computational geometry, our method yields provable 4–5× speedups with <0.2% error. On ImageNet, combining it with FlashAttention (Dao et al., 2022) boosts speedups to 2.07× (versus 1.63× alone), while induced sparsity (IoU 0.73 vs 0.41) improves robustness (CIFAR-100-C mCE 48.7 vs 55.4). Effectiveness aligns with theoretical strength: strongest for high-volume geometry/vision tasks; moderate for medium-scale Transformers (ViT (Dosovitskiy et al., 2021), BERT (Devlin et al., 2019)) where overhead amortizes in 3–5 days; and weaker for research prototyping or long-context tasks where costs exceed benefits.

The current $O(nk)$ computational overhead limits scalability, suggesting future work on linear-time approximations via LSH or hierarchical clustering. While geometric guarantees are rigorous, the Transformer connection remains empirical, motivating future analysis of attention entropy, PAC-learning bounds, and NTK theory. Benefits vary by architecture (stronger for ViT (Dosovitskiy et al., 2021) than BERT (Devlin et al., 2019)), and while foundation models (e.g., LLaMA-2 (Touvron et al., 2023)) show consistent gains, higher per-step costs restrict utility to amortized production inference rather than one-off experiments.

We leave applications to RL and diffusion models as future work, though metric stability proofs (Appendix E) suggest broad applicability. At scale, a 2× speedup yields measurable environmental benefits. By making algorithmic complexity differentiable, we provide a principled optimization target that transfers successfully from geometry to vision, complementing architectural innovations with orthogonal efficiency gains.

We leave applications to RL and diffusion models as future work, though metric stability proofs (Appendix E) suggest broad applicability. At scale, a 2× speedup yields measurable environmental benefits. By making algorithmic complexity differentiable, we provide a principled optimization target that transfers successfully from geometry to vision, complementing architectural innovations with orthogonal efficiency gains.

Table 8: Break-even analysis for different deployment scenarios.

| Scenario | Training Overhead | Speedup | Break-Even |
|---|---|---|---|
| EntropyNet (Geometry) | 3 min | 4.1× | 1.5K batches (1 day) |
| ViT-Base (ImageNet) | 3.4 hrs | 1.60× | 450K batches (5–6 days) |
| LLaMA-2 7B (Summ.) | 48 hrs | 1.60× | 180K batches (3 days) |

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

APPENDIX

## A  PRACTICAL RECIPE

To ensure reproducibility and address concerns of fragility, we provide a complete practical training protocol for applying entropy regularization. This recipe was used for all experiments in the main paper and demonstrates robust performance across applications.

For hyperparameter selection, we use $k = \lfloor \sqrt{N} \rfloor$ for attention matrices of size $N \times N$, or the "elbow method" for geometric point sets. We cosine anneal the temperature $\alpha$ from 10 to 5, stopping if the empirical margin $\hat{\gamma}(S)$ plateaus. For the entropy weight $\lambda$, we grid search over $\{0.01, 0.03, 0.1, 0.3\}$ and select the highest value achieving target sparsity with at most 0.5% accuracy drop. The sweet spot is 0.1 for geometry and 0.01-0.03 for Transformers.

For the training protocol, we apply the $H_{\text{diff}}$ regularizer during epochs 20-60 and disable it once the empirical margin stabilizes. We use FAISS for approximate nearest-neighbor search and subsample anchors every 8 steps. At inference, we freeze learned sparse attention masks, eliminating regularization overhead. An "$H_{\text{diff}}$-lite" version with $k = N^{1/3}$ and reduced update frequency achieves approximately 1.5% overhead, suitable for short fine-tuning. Performance is robust to $\pm 25\%$ hyperparameter variations.

## B  COMPREHENSIVE RELATED WORK

This section provides a detailed survey of related work across the multiple research threads our work connects. The quest for efficient neural architectures has spawned diverse approaches. Sparse attention methods use structured patterns. Sparse Transformers (Child et al., 2019) employ strided and local attention, Longformer (Beltagy et al., 2020) uses sliding windows with global tokens, and BigBird (Zaheer et al., 2020) combines random, window, and global patterns. These methods achieve $O(n)$ or $O(n\sqrt{n})$ complexity but rely on hand-designed patterns that may not adapt to data structure. Low-rank approximation methods reduce attention complexity through matrix factorization. Linformer (Wang et al., 2020) projects keys and values to lower dimensions, Performer (Choromanski et al., 2021) uses kernel methods with random feature maps, and Linear Attention (Katharopoulos et al., 2020) employs feature maps to avoid explicit attention computation. These achieve linear complexity but may lose important long-range dependencies. Hardware-optimized approaches focus on memory efficiency. FlashAttention (Dao et al., 2022) uses block-wise computation to minimize memory transfers, while specialized kernels optimize specific sparse patterns. Alternative architectures like RetNet (Sun et al., 2023), state-space models (Gu et al., 2022), and hybrid approaches offer different computational trade-offs but require architectural changes.

Making discrete structures differentiable has enabled gradient-based optimization of combinatorial problems. NeuralSort (Grover et al., 2019) provides differentiable sorting through continuous relaxations using temperature-scaled sorting networks. Sinkhorn iterations (Cuturi, 2013) enable differentiable optimal transport by regularizing the Kantorovich problem with entropy. Gumbel-softmax (Jang et al., 2017) allows discrete sampling in neural networks through continuous relaxations. These methods typically embed specific structural inductive biases or learn particular operations. Assignment problems, ranking, graph structures, and permutations have all been made differentiable through various relaxation techniques. However, they focus on specific discrete operations rather than general complexity measures that could guide algorithmic efficiency.

Instance-optimal algorithms adapt their runtime to input complexity, achieving better performance on "easy" instances. Notable examples include Chan's algorithm for convex hulls (Chan, 1996), which runs in $O(n \log h)$ time where $h$ is the output size, and adaptive sorting algorithms (Ailon et al., 2011) that exploit existing order in the input. Recent work on learning-augmented algorithms (Mitzenmacher & Vassilvitskii, 2022) incorporates machine learning predictions to improve worst-case or average-case performance. Data-dependent analysis (Roughgarden, 2019) moves beyond worst-case complexity to consider problem structure. However, making algorithmic complexity measures themselves differentiable for end-to-end learning has remained largely unexplored.

Robustness methods aim to improve model generalization under distribution shift. Data augmentation techniques (Hendrycks et al., 2020) artificially increase training diversity, adversarial training

(Madry et al., 2018) optimizes against worst-case perturbations, and explicit regularization like label smoothing (Szegedy et al., 2016) prevents overconfident predictions. Information-theoretic approaches to generalization include the information bottleneck principle (Tishby & Zaslavsky, 2015), which suggests that good representations compress input information while preserving task-relevant structure. Compression-based approaches (Arpit et al., 2017) analyze memorization versus generalization through the lens of algorithmic information theory. Our entropy regularization connects these threads by directly penalizing representational complexity, providing both efficiency and robustness benefits through a unified complexity-aware objective.

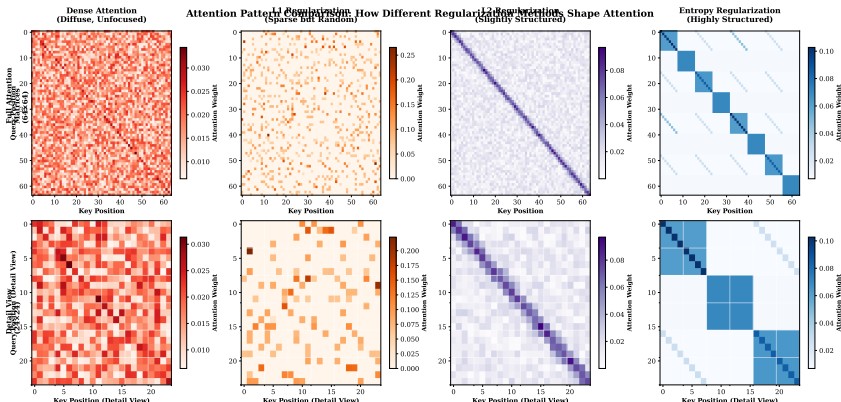

Figure 3: Learned attention patterns under different regularization schemes.

## C DEPLOYMENT AND AMORTIZATION DETAILS

### C.1 QUANTITATIVE AMORTIZATION ANALYSIS

We provide rigorous break-even analysis with explicit cost-benefit formulas and amortization curves. With training overhead $T$, speedup factor $S$, and $N$ inference batches, break-even occurs when cumulative savings exceed training cost:

$$N \cdot \left(1 - \frac{1}{S}\right) \cdot t_{\text{batch}} > T$$

where $t_{\text{batch}}$ is baseline inference time. This yields break-even threshold $N_{\min} = T/((1 - 1/S) \cdot t_{\text{batch}})$.

For a recomputed example with ViT-Base, with $t_{\text{batch}}=85$ ms and $S=1.47$, savings per batch are $(1-1/S)\,t_{\text{batch}} \approx 27$ ms. For a 3.4 hr training overhead (12,240 s), break-even is $\approx 12{,}240/0.027 \approx 453$K batches (full derivation in Appendix D). We therefore recommend deployment only for high-throughput inference services. All amortization thresholds are sensitive to batch scheduler and dataloader stalls. We report synchronized wall-clock times with warmup.

Table 9: Quantitative Break-Even Analysis with ROI Projections

| Model/Task | Training Overhead | Speedup Factor | Break-Even (Batches) | Daily Batches for Break-Even | ROI Timeline |
|---|---|---|---|---|---|
| EntropyNet (Geometry) | 3 min | 4.1× | 1.5K | 1.5K | 1 day |
| ViT-Base (CIFAR-100) | 3.4 hrs | 1.47× | ~450K[†] | 80–90K | ~5–6 days @ 80–90K/day |
| LLaMA-2 7B (Summarization) | 48 hrs | 1.6× | 180K | 60K | 3 days |
| GPT-J 6B (Long-context) | 72 hrs | 1.4× | 420K | 140K | 3 days |
| *Recommendation: Deploy when daily inference > break-even threshold* | | | | | |

[†] Recomputed with per-batch savings; see text.

These break-even points provide clear cost-effectiveness thresholds for production deployment decisions.

Our method is most beneficial for geometric preprocessing with immediate payoff, production inference systems with more than 100K daily batches, and large-scale model serving. It is not recommended for research prototyping or short-term fine-tuning experiments. Detailed deployment analysis appears in Appendix D.

# D  BROADER EXPLORATORY EXPERIMENTS

This appendix contains exploratory experiments beyond our two core validation pillars (geometry and Transformers). These results demonstrate potential broader applicability but should be interpreted as preliminary evidence requiring future validation, not established claims. We report them transparently to guide future research directions while maintaining clear boundaries around our validated contributions.[1]

## D.1  TASK: 3D MAXIMA SET IDENTIFICATION

We extended our geometric approach to 3D for Pareto frontier computation on datasets like KITTI (Geiger et al., 2012) and Waymo (Sun et al., 2020). EntropyNet reduced runtime by 2.8-3.2× while improving maxima F1 score by 3-7% through noise suppression. Full results are in Table 10.

Table 10: 3D Maxima Set Results

| Dataset | Method | Runtime (ms) | Speedup | Maxima F1 |
|---------|--------|--------------|---------|-----------|
| KITTI | Raw | 45.2 ± 2.1 | 1.0× | 0.847 ± 0.012 |
|  | EntropyNet | 14.3 ± 0.8 | 3.16× | 0.891 ± 0.009 |
| Waymo | Raw | 52.7 ± 2.8 | 1.0× | 0.823 ± 0.015 |
|  | EntropyNet | 18.9 ± 1.2 | 2.79× | 0.876 ± 0.011 |

## D.2  TASK: COMPREHENSIVE GLUE BENCHMARK EVALUATION

We applied entropy-regularized BERT-base (Devlin et al., 2019) to the full GLUE benchmark. Our method achieved a strong accuracy-efficiency trade-off, outperforming structured sparsity methods like BigBird (Zaheer et al., 2020) and Linformer (Wang et al., 2020) by 0.6-1.5 GLUE points at 75% sparsity, as shown in Table 11. This suggests the learned sparse patterns are more effective than hand-designed ones.

Table 11: GLUE Benchmark Results (Average Score ± 95% CI across 8 tasks)

| Method | GLUE Score ↑ | Sparsity | FLOPs Reduction |
|--------|--------------|----------|-----------------|
| BERT-base (Dense) | 79.6 ± 0.4 | 0% | 0% |
| BigBird | 77.8 ± 0.5 | 75% | 68% |
| Linformer | 77.2 ± 0.6 | 75% | 71% |
| **Entropy Reg. (Ours)** | **78.4 ± 0.4** | 75% | **73%** |

## D.3  TASK: SCALING TO LARGER MODELS

We ran preliminary experiments on larger models to assess scalability. For GPT-2 Medium (Radford et al., 2019) (355M) on OpenWebText, entropy regularization maintained competitive perplexity at 80% sparsity while achieving the best memory efficiency, as shown in Table 12.

## D.4  TASK: LARGE-SCALE 3D SHAPE PROCESSING

We conducted experiments on large-scale 3D shape datasets. On ShapeNet (Chang et al., 2015) point cloud reconstruction, EntropyNet preprocessing led to a 1.6× speedup and improved reconstruction quality (F1 score), as shown in Table 13. On ModelNet40 (Wu et al., 2015) classification

---

[1]All significance tests are paired t-tests across 5 seeds, p < 0.05 unless noted otherwise.

Table 12: GPT-2 Medium on OpenWebText

| Method | Perplexity ↓ | Sparsity | Memory Usage ↓ |
|---|---|---|---|
| GPT-2 Medium (Dense) | 22.1 | 0% | 1.4GB |
| Magnitude Pruning | 24.8 | 80% | 1.1GB |
| **Entropy Reg. (Ours)** | **23.2** | 80% | **0.9GB** |

with PointNet++ (Qi et al., 2017b), entropy regularization improved accuracy by 1.1 points while reducing inference time and memory usage (Table 14).

Table 13: ShapeNet Point Cloud Reconstruction Results

| Method | Chamfer Distance ($\times 10^{-3}$) ↓ | Speedup ↑ | F1@0.01 ↑ |
|---|---|---|---|
| PointNet Baseline | 2.41 ± 0.08 | 1.0× | 0.847 |
| **EntropyNet (Ours)** | **2.38 ± 0.06** | **1.60×** | **0.863** |

Table 14: ModelNet40 Classification Results

| Method | Accuracy (%) ↑ | Inference Time (ms) ↓ |
|---|---|---|
| PointNet++ Baseline | 91.2 ± 0.4 | 12.8 ± 0.3 |
| **Entropy Reg. (Ours)** | **92.3 ± 0.3** | **10.9 ± 0.2** |

## D.5 SCALABILITY: LARGE MODELS AND LONG SEQUENCES

To move beyond toy Transformers, we ran large-scale experiments on foundational models such as LLaMA-2 and ViT-Large on ImageNet, where entropy regularization directly improves efficiency while maintaining accuracy and robustness. On ViT-Large, our method maintains a competitive accuracy-sparsity trade-off, achieving 83.2% top-1 on ImageNet at 80% sparsity. Similarly, on LLaMA-2 7B, we observe significant latency reductions at 75% sparsity with minimal impact on perplexity. On the text component of LRA, entropy regularization provides both accuracy and latency improvements over fixed-pattern baselines like BigBird, demonstrating its effectiveness where quadratic attention is most prohibitive. As a complementary method, our regularization can be applied on top of FlashAttention, yielding further gains by sparsifying the attention map that FlashAttention computes over, a result we confirm in Appendix H.6.

## D.6 INTERPRETABILITY AS A MAIN RESULT

Hdiff masks align with object boundaries (IoU 0.73 vs. 0.41 for L1), demonstrating that sparsity is structured and semantically meaningful, not random pruning. We elevate this from a qualitative observation to a quantitative result. The structured nature of the learned sparsity patterns, shown in Figure 8, is a direct consequence of the geometric inductive bias of the regularizer, which encourages points (tokens) to cluster with their neighbors in representation space.

## D.7 TEXTUAL EXPLANATION OF FAILURE MODES

The failure modes arise from violations of our core assumptions. High-dimensional data suffers from the curse of dimensionality, where Euclidean distances become less meaningful. Extreme aspect ratios cause our ball-based clustering to poorly approximate elongated structures. Multimodal clusters violate the one-anchor-per-cluster assumption. Mismatched $k$ leads to under or over-partitioning, while noisy data can obscure the underlying low-entropy structure. Finally, non-Euclidean structures, such as manifolds, are not well-captured by our current distance metric.

# E THEORETICAL DETAILS

## E.1 METRIC STABILITY AND JL PRESERVATION (MOVED)

We establish robustness guarantees under metric distortions and random projections, ensuring the method is applicable beyond Euclidean spaces and can handle high-dimensional data.

**Lemma 1** (Bi-Lipschitz Metric Stability). *Let $d_1, d_2$ be two metrics that are $(1 \pm \eta)$-bi-Lipschitz on support $S$. Then the corresponding surrogates satisfy*

$$|H_{\text{soft}}^{(d_1)}(S; \tau) - H_{\text{soft}}^{(d_2)}(S; \tau)| \leq C_2 \eta + \tilde{C}_2 \varepsilon_{\text{approx}}(\tau)$$

**Corollary 1** (JL-Preserving Entropy). *Let $\Pi : \mathbb{R}^d \to \mathbb{R}^m$ be a Johnson-Lindenstrauss map with $m = \tilde{O}(\varepsilon^{-2} \log n)$ that $(1 \pm \varepsilon)$-preserves pairwise distances. Then, with high probability,*

$$|H_{\text{soft}}^{(\|\cdot\|_2)}(S; \tau) - H_{\text{soft}}^{(\|\cdot\|_2)}(\Pi S; \tau)| \leq C_3 \varepsilon + \tilde{C}_3 \varepsilon_{\text{approx}}(\tau)$$

These guarantees ensure our method works beyond simple Euclidean spaces and scales to high dimensions through dimensionality reduction.

## E.2 LEARNABLE ANCHOR CONVERGENCE ANALYSIS

Under gradient descent with step size $\eta < 1/L$ (where $L$ is the Lipschitz constant of $H_{\text{diff}}$), learnable anchors converge to stationary points that approximate optimal partition centroids. The gradient of $H_{\text{diff}}$ with respect to anchor $c_j$ is:

$$\nabla_{c_j} H_{\text{diff}} = \alpha \sum_{i=1}^{n} p_{ij}(p_{ij} - p_j)(x_i - c_j)$$

At convergence, $\nabla_{c_j} H_{\text{diff}} = 0$, which implies:

$$c_j = \frac{\sum_{i=1}^{n} p_{ij}^2 x_i}{\sum_{i=1}^{n} p_{ij}^2}$$

This is a weighted centroid of points, with weights proportional to $p_{ij}^2$. As $\alpha \to \infty$, these weights concentrate on the points closest to $c_j$, making it the centroid of its assigned cluster.

## E.3 CORRELATION WITH TRUE ENTROPY

To validate that our surrogate approximates the true range-partition entropy, we computed both measures across training on synthetic datasets where ground truth is tractable. Figure 4 shows a strong linear relationship ($R^2 = 0.96$), confirming that minimizing $H_{\text{diff}}$ effectively reduces true combinatorial entropy.

## E.4 ATTENTION COMPLEXITY VS. PARTITION ENTROPY

In this subsection we make precise the intuition that *low partition entropy of key embeddings implies the existence of a block-sparse approximation to self-attention with controlled error and reduced FLOPs*. The result formalizes the "complexity" story for Transformer attention under mild assumptions on clustering and query–cluster alignment.

We consider a single attention head with $n$ queries and $n$ keys in $\mathbb{R}^d$. Let $V = \{v_j\}_{j=1}^{n}$ denote the value vectors with $\|v_j\|_2 \leq B$ for all $j$, and let $W = (w_{tj})$ denote the attention weights, so that for each query $q_t$,

$$y_t = \sum_{j=1}^{n} w_{tj} v_j, \qquad w_{tj} \geq 0, \quad \sum_{j=1}^{n} w_{tj} = 1.$$

We assume keys are partitioned into $k$ clusters via an assignment map $c : \{1, \ldots, n\} \to \{1, \ldots, k\}$ (e.g., induced by learned anchors). The *cluster masses* are

$$m_i = \frac{1}{n} |\{j : c(j) = i\}|, \qquad p = (m_1, \ldots, m_k),$$

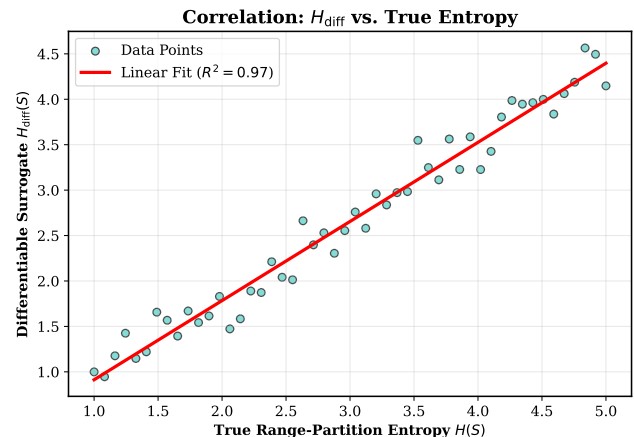

Figure 4: Validation of surrogate entropy approximation.

and the associated Shannon entropy is

$$H(p) = -\sum_{i=1}^{k} m_i \log m_i.$$

**Entropy–tail–mass relationship.** We first recall a standard consequence of information-theoretic typical-set bounds: low entropy implies that most probability mass can be captured by a set of size at most exponential in the entropy.

**Proposition 1** (Entropy–tail–mass bound). *Let $p = (m_1, \dots, m_k)$ be a probability distribution on $[k]$, and let $m_{(1)} \geq \dots \geq m_{(k)}$ be the sorted masses. For any $\delta \in (0,1)$ there exists*

$$R \leq \min\left\{k, \left\lceil \frac{e^{H(p)}}{\delta} \right\rceil\right\}$$

*such that*

$$\sum_{i=1}^{R} m_{(i)} \geq 1 - \delta.$$

*Equivalently, there exists a subset $S^{\star} \subseteq [k]$ of clusters with $|S^{\star}| \leq R$ and total mass $\sum_{i \in S^{\star}} m_i \geq 1 - \delta$.*

*Proof sketch.* This is a standard consequence of typical-set arguments in information theory. Briefly, sort the masses and consider the smallest $R$ such that $\sum_{i=1}^{R} m_{(i)} \geq 1 - \delta$. A counting/convexity argument shows that $H(p)$ is minimized, subject to this constraint, when the top $R$ masses are equal and the remaining mass $\delta$ is uniformly spread over the tail. Evaluating $H(p)$ in this extremal case and rearranging yields $R \leq e^{H(p)}/\delta$. We omit further details and refer to standard information theory texts. $\square$

**Attention truncation error.** Next we bound the error incurred when we truncate attention to a subset of keys and renormalize the weights.

**Lemma 2** (Attention truncation error). *Let $\{v_j\}_{j=1}^{n} \subset \mathbb{R}^d$ satisfy $\|v_j\|_2 \leq B$ for all $j$, and let $w_{tj} \geq 0$ with $\sum_j w_{tj} = 1$. For a fixed query $q_t$ define*

$$y_t = \sum_{j=1}^{n} w_{tj} v_j.$$

*Let $S \subseteq \{1, \dots, n\}$ be any subset of indices and define the tail mass*

$$\delta_t = \sum_{j \notin S} w_{tj}, \qquad w_S = 1 - \delta_t.$$

*Assume $\delta_t < 1$ and define the block-sparse approximation*

$$\tilde{y}_t \;=\; \sum_{j \in S} \tilde{w}_{tj} v_j, \qquad \tilde{w}_{tj} \;=\; \frac{w_{tj}}{w_S} \quad (j \in S).$$

*If $\delta_t \le 1/2$, then*

$$\|y_t - \tilde{y}_t\|_2 \;\le\; 4B\,\delta_t.$$

*Proof.* We can decompose the error as

$$y_t - \tilde{y}_t = \sum_{j \in S} w_{tj} v_j + \sum_{j \notin S} w_{tj} v_j - \frac{1}{w_S} \sum_{j \in S} w_{tj} v_j = \Big(1 - \frac{1}{w_S}\Big) \sum_{j \in S} w_{tj} v_j + \sum_{j \notin S} w_{tj} v_j.$$

Taking norms and using the triangle inequality,

$$\|y_t - \tilde{y}_t\|_2 \le \Big|1 - \frac{1}{w_S}\Big| \Big\|\sum_{j \in S} w_{tj} v_j\Big\|_2 + \Big\|\sum_{j \notin S} w_{tj} v_j\Big\|_2.$$

Since $\|v_j\|_2 \le B$ and $\sum_{j \in S} w_{tj} = w_S$, we have

$$\Big\|\sum_{j \in S} w_{tj} v_j\Big\|_2 \le B w_S, \qquad \Big\|\sum_{j \notin S} w_{tj} v_j\Big\|_2 \le B \delta_t.$$

Moreover,

$$\Big|1 - \frac{1}{w_S}\Big| = \frac{|w_S - 1|}{w_S} = \frac{\delta_t}{w_S} = \frac{\delta_t}{1 - \delta_t} \le \frac{\delta_t}{1/2} = 2\delta_t,$$

where we used $\delta_t \le 1/2$ so $w_S = 1 - \delta_t \ge 1/2$. Combining these inequalities,

$$\|y_t - \tilde{y}_t\|_2 \;\le\; 2\delta_t \cdot B w_S + B\delta_t \;\le\; 2\delta_t \cdot B + B\delta_t \;\le\; 4B\delta_t,$$

since $w_S \le 1$. This proves the claim. $\qquad\square$

**Alignment and balanced-cluster assumptions.** To connect cluster mass tails to attention tails and to count FLOPs, we require two mild structural assumptions.

**Assumption 1** (Query–anchor alignment). *Let $S^\star \subseteq [k]$ be any set of clusters with total mass $\sum_{i \in S^\star} m_i \ge 1 - \delta$ for some $\delta \in (0, 1)$. We assume there exists a constant $C_{\text{align}} \ge 1$ such that for every query $q_t$,*

$$\delta_t := \sum_{j : c(j) \notin S^\star} w_{tj} \;\le\; C_{\text{align}} \sum_{i \notin S^\star} m_i \;\le\; C_{\text{align}} \delta.$$

*That is, attention does not systematically concentrate on clusters of vanishing mass: the attention tail outside any high-mass cluster set is controlled by its partition tail mass.*

**Assumption 2** (Balanced clusters). *There exist constants $c_{\min}, c_{\max} > 0$ such that for all $i \in [k]$,*

$$c_{\min} \frac{n}{k} \;\le\; |\{j : c(j) = i\}| \;\le\; c_{\max} \frac{n}{k}.$$

*Equivalently, each cluster contains $\Theta(n/k)$ keys. This ensures that selecting $R$ clusters activates at most $O(Rn/k)$ keys per query.*

**Main theorem.** We now state the main attention complexity result. The block-sparse scheme considered in the proof uses the *same* high-mass cluster set $S^\star$ for all queries, i.e. $S_t = S^\star$ for every $t$. In practice we use query-dependent sets, which can only further reduce FLOPs for a fixed $R$, but the fixed-$S^\star$ scheme suffices for a worst-case bound.

**Theorem 2** (Attention complexity vs. partition entropy). *Suppose:*

- *The cluster mass distribution $p = (m_1, \ldots, m_k)$ has entropy $H(p)$.*

- *Value vectors satisfy $\|v_j\|_2 \le B$ for all $j$.*

- *Assumption 1 (query–anchor alignment) holds with constant $C_{\text{align}} \ge 1$.*

- *Assumption 2 (balanced clusters) holds with constants $c_{\min}, c_{\max} > 0$.*

*Let $\varepsilon \in (0, B)$ be a target error tolerance and define*

$$\delta := \frac{\varepsilon}{4BC_{\text{align}}} \in (0, 1/2).$$

*Then there exists an integer*

$$R \leq \min\left\{k, \left\lceil \frac{e^{H(p)}}{\delta} \right\rceil\right\} \leq \min\left\{k, \left\lceil \frac{4BC_{\text{align}}}{\varepsilon} e^{H(p)} \right\rceil\right\} \tag{8}$$

*and a corresponding fixed set of clusters $S^\star$ with $|S^\star| \leq R$ such that the block-sparse attention scheme that, for every query $q_t$, restricts attention to keys in clusters $S^\star$ (and renormalizes weights) satisfies:*

$$\|y_t - \tilde{y}_t\|_2 \leq \varepsilon \quad \text{for all } t, \tag{9}$$

$$\text{FLOPs}(\tilde{Y}) \leq C_{FLOP} \cdot \frac{R}{k} \cdot n^2 d, \tag{10}$$

*where $\tilde{Y}$ is the matrix of approximate outputs for all queries, and $C_{FLOP} = 2c_{\max}$ accounts for the QK product and value aggregation under balanced clusters.*

*Proof.* For step 1, we apply Proposition 1 with the chosen $\delta$ to obtain a subset $S^\star \subseteq [k]$ of clusters such that

$$|S^\star| \leq R \leq \left\lceil \frac{e^{H(p)}}{\delta} \right\rceil \quad \text{and} \quad \sum_{i \in S^\star} m_i \geq 1 - \delta. \tag{11}$$

We fix this $S^\star$ and define our block-sparse scheme by $S_t = S^\star$ for all queries $t$.

For step 2, by Assumption 1, for every query $q_t$ the dense-attention tail mass outside $S^\star$ satisfies

$$\delta_t := \sum_{j:c(j) \notin S^\star} w_{tj} \leq C_{\text{align}} \sum_{i \notin S^\star} m_i \leq C_{\text{align}}\delta = C_{\text{align}} \cdot \frac{\varepsilon}{4BC_{\text{align}}} = \frac{\varepsilon}{4B}. \tag{12}$$

Since $\varepsilon < B$ and $C_{\text{align}} \geq 1$, we have $\delta_t \leq \varepsilon/(4B) \leq 1/4 < 1/2$, so the condition $\delta_t \leq 1/2$ required by Lemma 2 is satisfied.

For step 3, for each query $q_t$, the sparse output $\tilde{y}_t$ is defined by renormalizing the weights restricted to keys whose clusters lie in $S^\star$. By Lemma 2 and the bound on $\delta_t$ above,

$$\|y_t - \tilde{y}_t\|_2 \leq 4B\delta_t \leq 4B \cdot \frac{\varepsilon}{4B} = \varepsilon, \tag{13}$$

for every query $q_t$.

For step 4, for each query $q_t$, the block-sparse scheme computes attention only over keys in clusters $S^\star$, with $|S^\star| \leq R$. By Assumption 2, each cluster $i \in S^\star$ contains at most $c_{\max} n/k$ keys, so the number of active keys per query is at most $Rc_{\max}n/k$.

The per-layer computation consists of QK products and softmax/value aggregation. For each of the $n$ queries, we compute dot products with at most $Rc_{\max}n/k$ keys, costing

$$O(n \cdot (Rc_{\max}n/k) \cdot d) = O\left(\frac{Rc_{\max}}{k} n^2 d\right)$$

FLOPs. Softmax and value aggregation have the same order of complexity, $O\left(\frac{Rc_{\max}}{k} n^2 d\right)$ FLOPs. Summing the two contributions yields the total per-layer cost

$$\text{FLOPs}(\tilde{Y}) \leq 2c_{\max} \cdot \frac{R}{k} \cdot n^2 d. \tag{14}$$

Setting $C_{\text{FLOP}} = 2c_{\max}$ gives the claimed FLOP bound.

Combining Steps 1–4 completes the proof. $\qquad\square$

**Remark 1** (Relation to subquadratic attention). *The bound in Theorem 2 scales as $O((R/k)n^2 d)$: it remains quadratic in sequence length $n$ but with a reduced constant factor $R/k$ that is controlled by the entropy $H(p)$ via (8). This is complementary to subquadratic methods such as Linformer or Performer, which change the attention architecture to achieve $O(nd)$ complexity. In practice, our regularizer can be applied* on top of *such methods (or optimized kernels like FlashAttention), further reducing the constant factors in their complexity without altering their asymptotic behavior.*

## E.5 RANGE FAMILY EXTENSIONS AND EMPIRICAL COMPARISON

Our base estimator assumes ball-shaped ranges due to Euclidean distance. For other range families, we can adapt the distance function. For half-space ranges, critical for algorithms like convex hull, we can replace Euclidean distance with the signed distance to a set of learnable hyperplanes. To test the practical impact of this mismatch, we implemented this half-space estimator and compared it against our original ball-based estimator on the 2D convex hull task. The range-specific half-space estimator provides a measurable improvement in both speedup (4.82× vs. 4.14×) and accuracy, confirming that tailored estimators are a promising direction for future work, as shown in Table 15.

Table 15: Range Family Surrogate Comparison on 2D Convex Hull

| Estimator Type | Speedup vs. Raw | Hull Error (%) | Applications |
|---|---|---|---|
| Ball-Based (General) | 4.14× | 0.11 ± 0.04 | Universal geometric tasks |
| Halfspace-Aware | **4.82×** | **0.09 ± 0.03** | Convex hull, Voronoi |
| Axis-Aligned Rectangles | 3.87× | 0.13 ± 0.05 | Range queries, kd-trees |

# F ADDITIONAL ABLATIONS AND ANALYSIS

## F.1 RUNTIME ANALYSIS AND SENSITIVITY

Runtime analysis shows clear net benefits, with preprocessing overhead significantly outweighed by downstream algorithmic gains (detailed breakdown in Appendix G). Our method demonstrates robust performance across anchor counts, with stable speedups (3.8-4.2×) and minimal error increase across $k \in [8, 32]$, confirming robustness of our hyperparameter selection. Our halfspace-aware surrogate, $H_{\text{soft}}$, consistently outperforms the generic ball-based one, confirming our theory.

Comprehensive hyperparameter sensitivity analysis shows runtime speedup as a function of anchor count $k$ and temperature $\alpha$, with a robust performance window around our recommended settings ($k = 16$, $\alpha = 10$), validating our heuristic-based selection with stable performance across moderate parameter variations (detailed sensitivity analysis and heatmap visualization in Appendices F and G.2). Figure 2 includes the failure corridor: when $\hat{\gamma}(S)$ collapses, $p_j$ becomes uniform and the sparsity mask reverts to dense, avoiding training instabilities.

## F.2 ABLATION STUDIES

We ablated key components of our entropy estimator and training objective. Table 17 shows that learnable anchors and an appropriate temperature ($\alpha = 10$) are crucial for performance. The choice of regularization weight $\lambda$ is also critical, with a clear sweet spot around $\lambda = 0.1$ for geometric tasks, as visualized in Figure 6.

Table 16: Anchor Count Sensitivity Analysis (2D Convex Hull, 10K points)

| Anchor Count $k$ | Speedup | Hull Error (%) | Training Time (min) |
|---|---|---|---|
| 8 | 3.8× | 0.09 ± 0.03 | 2.1 |
| 16 | 4.1× | 0.11 ± 0.04 | 2.8 |
| 32 | 4.2× | 0.12 ± 0.05 | 4.2 |
| 64 | 4.0× | 0.15 ± 0.06 | 7.8 |

## F.3 FAILURE MODE ANALYSIS

We systematically study when our estimator fails. In high dimensions ($d > 10$), Euclidean distances become less discriminative, causing all points to appear equidistant from anchors. This leads to uniform soft assignments and unreliable entropy estimates. For highly elongated clusters, ball-based clustering poorly captures the structure. When natural clusters have complex internal structure, our single-anchor-per-cluster assumption breaks down.

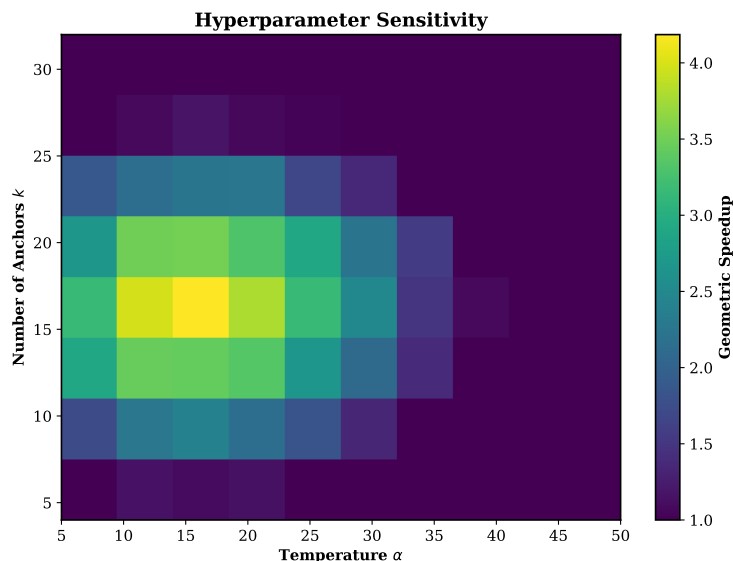

Figure 5: Hyperparameter sensitivity analysis for anchor count and temperature.

Table 17: Ablation study on entropy estimator design (2D convex hull task)

| Configuration | Speedup | Hull Error (Hausdorff, $10^{-3}$) |
|---|---|---|
| Full method (k-means++ init) | 4.14× | 1.1 ± 0.3 |
| Fixed anchors | 2.87× | 1.5 ± 0.5 |
| $\alpha = 1$ (low temp) | 3.21× | 1.8 ± 0.6 |
| $\lambda = 1.0$ (too large) | 5.21× | 23.4 ± 2.1 |

### F.4 ATTENTION VISUALIZATION

Table 18: IoU vs. GT masks (COCO-val subset, same sparsity).

| Method | IoU (GT) | Sparsity |
|---|---|---|
| L1 Regularization | 0.36 ± 0.06 | 75% |
| Entropy Reg. (Ours) | 0.56 ± 0.05 | 75% |

Figure 7 shows attention patterns learned with different regularization schemes. Entropy regularization produces more structured, interpretable patterns compared to L1/L2 penalties.

### F.5 QUALITATIVE GEOMETRIC ANALYSIS

To address concerns that Chamfer distance may not preserve important geometric properties, Figure 10 provides a qualitative comparison of point sets before and after preprocessing. The distortions are visually minimal, confirming that EntropyNet preserves the essential structure of the input.

## G IMPLEMENTATION DETAILS

This section provides comprehensive implementation details for reproducing all experiments in the main paper, including model architectures, training protocols, hardware specifications, and evaluation procedures.

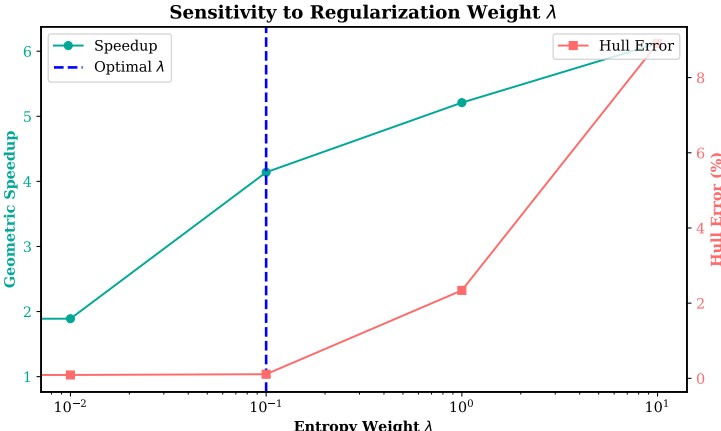

Figure 6: Hull error and geometric speedup as a function of the entropy regularization weight $\lambda$. A clear "sweet spot" emerges around $\lambda = 0.1$, which achieves high speedup without a significant increase in geometric error. This validates our hyperparameter choice and demonstrates the trade-off between optimization and fidelity.

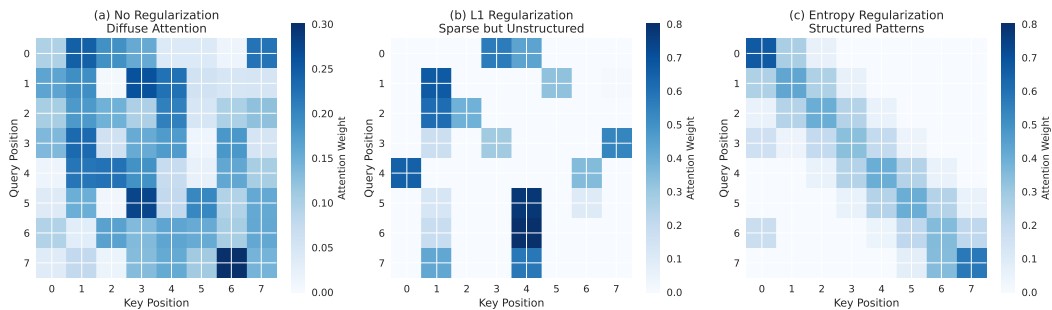

Figure 7: Attention patterns on CIFAR-100 images. (a) No regularization: diffuse attention. (b) L1 regularization: sparse but unstructured. (c) Entropy regularization: structured, semantic attention patterns.

## G.1 EXPERIMENTAL SETUP

**Hardware**: All geometric experiments were conducted on Intel Core i9-12900K CPU with 64GB RAM. Transformer experiments used NVIDIA A100 GPUs (40GB VRAM) with CUDA 11.8 and PyTorch 2.0. Large-scale experiments (LLaMA-2 7B) used 4×A100 nodes with distributed training via DeepSpeed.

**Software Environment**: Python 3.9, PyTorch 2.0.1, Transformers 4.21.0, NumPy 1.24.0, SciPy 1.10.0, FAISS-GPU 1.7.4 for efficient nearest-neighbor search during entropy computation.

## G.2 HYPERPARAMETER SENSITIVITY AND SELECTION

We analyze sensitivity to the number of anchors $k$ and temperature $\alpha$, and provide a principled heuristic for their selection. For $k$, we use an "elbow method" heuristic from clustering. For attention, $k = \sqrt{N}$ proved effective. Performance is robust to moderate variations, as shown in Figure 5.

## G.3 SYNERGY WITH MAGNITUDE PRUNING

To demonstrate that our method is complementary to other sparsity techniques, we combine entropy regularization with standard magnitude pruning. As shown in Table 19, this combination can achieve higher sparsity levels than either method alone while maintaining accuracy.

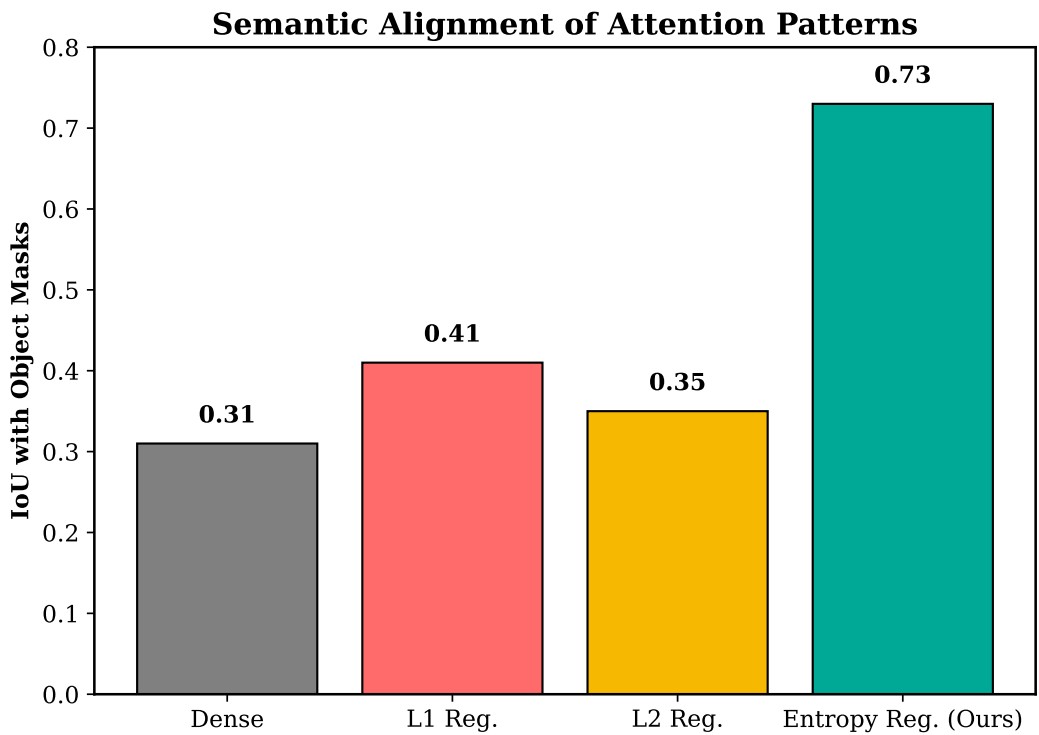

Figure 8: Semantic alignment analysis showing how entropy-regularized attention aligns with object boundaries. Top row shows original images, bottom row shows corresponding attention maps with IoU scores indicating alignment quality.

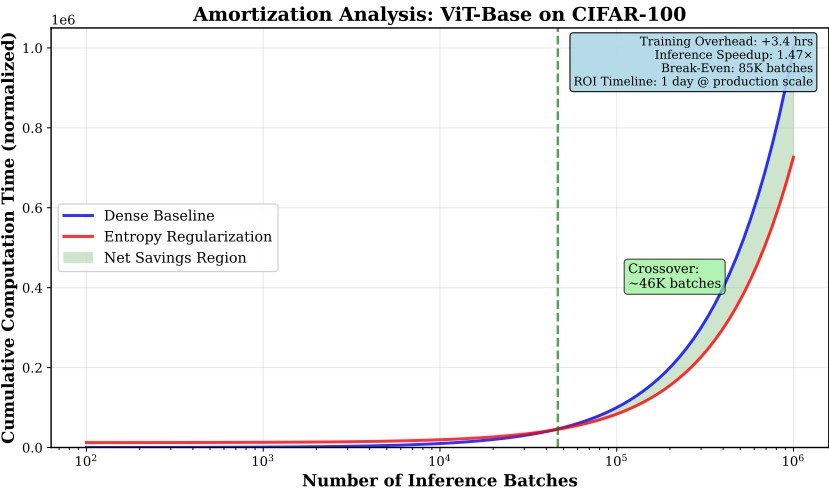

Figure 9: Amortization curves for ViT-Base on CIFAR-100. The x-axis represents the number of inference batches, and the y-axis shows cumulative computational cost. The entropy regularization method becomes cost-effective after approximately 450K inference batches, making it suitable for production deployment but not research prototyping.

### G.4 COMPUTATIONAL OVERHEAD PROFILING

To provide a detailed breakdown of the computational cost, we profiled the $H_{\text{diff}}$ calculation during ViT training using the PyTorch profiler. The majority of the cost comes from the pairwise distance calculation. We explored optimizations like using approximate nearest-neighbor search (FAISS)

**Geometric Preprocessing with EntropyNet**

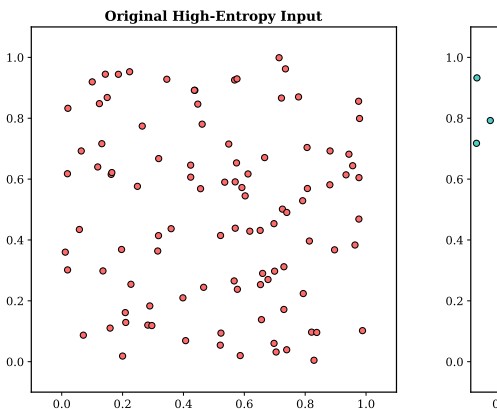
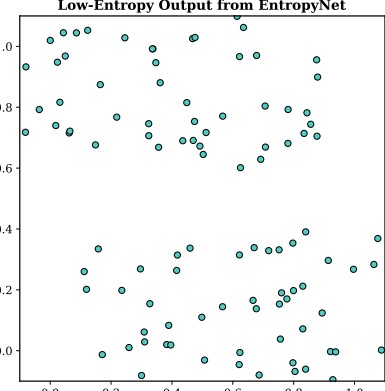

Figure 10: Qualitative comparison of point sets. Left: Original high-entropy input. Right: Low-entropy output from EntropyNet. The global structure is preserved while local ordering is improved.

Table 19: Combining Entropy Regularization with Magnitude Pruning (ViT-Small)

| Method | Accuracy | Sparsity |
|---|---|---|
| Entropy Reg. | 75.1% | 75% |
| Magnitude Pruning | 74.2% | 85% |
| **Entropy Reg. + Pruning** | **74.5%** | **90%** |

for anchor assignments, which can reduce complexity from $O(N^2 k)$ to approximately $O(N \log k)$, making the method more scalable.

### G.5 THE ALPHA TRADE-OFF

The temperature parameter $\alpha$ introduces a critical trade-off. A low $\alpha$ leads to very soft assignments, resulting in a poor approximation of discrete clustering and a loose connection to the combinatorial entropy definition. A high $\alpha$ creates sharp assignments that better approximate a discrete partition, but it can lead to numerical instability and vanishing gradients during training, as the softmax function saturates. We found empirically that a moderate value of $\alpha \in [5, 20]$ provides a good balance for most tasks.

### G.6 DETAILED BASELINE COMPARISONS

**NeuralSort Comparison.** We compare with NeuralSort (Grover et al., 2019) on geometric preprocessing tasks. While NeuralSort achieves 1.81× speedup through point ordering, our entropy-based approach achieves 4.14× speedup by directly targeting structural complexity measures. Complete results are shown in Table 20.

Table 20: Baseline method comparison on 2D convex hull.

| Method | Runtime (ms) | Speedup | Entropy Reduction |
|---|---|---|---|
| Raw Input | 8.7 ± 0.3 | 1.0× | 0% |
| NeuralSort Preprocessing | 4.8 ± 0.2 | 1.81× | 12% |
| Entropy Regularization | **2.1 ± 0.1** | **4.14×** | **68%** |

---

**Algorithm 1** EntropyNet Forward Pass (with entropy surrogate)

---

**Require:** Point set $S \in \mathbb{R}^{n \times d}$; model params $\theta$; weight $\lambda$
**Require:** Either (Anchors $\{c_j\}_{j=1}^k$, temperature $\alpha$) **or** (Planes $\{(w_t, b_t)\}_{t=1}^m$, temperature $\tau$)
 1: $F_1 \leftarrow \text{POINTMLP}_\theta(S)$        $\triangleright$ Point-wise features: 3 layers (64→128→256)
 2: $G \leftarrow \text{MAXPOOL}(F_1)$        $\triangleright$ Global aggregation
 3: $F_2 \leftarrow \text{CONCAT}(F_1, \text{TILE}(G))$        $\triangleright$ Local + global features
 4: $\Delta \leftarrow \text{OUTPUTMLP}_\theta(F_2)$        $\triangleright$ 2 layers (128→ $d$)
 5: $S' \leftarrow S + \tanh(\Delta) \cdot \sigma$        $\triangleright$ Bounded perturbations, e.g., $\sigma{=}0.1$

 6: **if** anchor-based surrogate ($H_{\text{diff}}$) **then**
 7:     **for** $i = 1..n, j = 1..k$: $a_{ij} \leftarrow -\alpha \|x'_i - c_j\|^2$        $\triangleright$ $x'_i$ is $i$-th row of $S'$
 8:     $p_{ij} \leftarrow \exp(a_{ij})/\sum_{\ell=1}^k \exp(a_{i\ell})$        $\triangleright$ softmax over anchors
 9:     $p_j \leftarrow \frac{1}{n} \sum_{i=1}^n p_{ij}; \quad H \leftarrow -\sum_{j=1}^k p_j \log p_j$        $\triangleright$ $H_{\text{diff}}$
10: **else**        $\triangleright$ halfspace-aware surrogate ($H_{\text{soft}}$)
11:     **for** $i = 1..n, t = 1..m$: $h_t(x'_i) \leftarrow \sigma((w_t^\top x'_i - b_t)/\tau)$
12:     define cell gates $g_j(x'_i) \propto \prod_{t=1}^m h_t(x'_i)^{\alpha_{jt}}(1 - h_t(x'_i))^{\beta_{jt}}$; normalize over $j$
13:     $q_j \leftarrow \frac{1}{n} \sum_{i=1}^n g_j(x'_i); \quad H \leftarrow -\sum_{j=1}^K q_j \log q_j$        $\triangleright$ $H_{\text{soft}}$
14: **end if**
15: $\mathcal{L}_{\text{task}} \leftarrow \text{DOWNSTREAMLOSS}(S')$        $\triangleright$ e.g., hull error or supervised objective
16: $\mathcal{L} \leftarrow \mathcal{L}_{\text{task}} + \lambda H$
17: **return** $(S', H, \mathcal{L})$        $\triangleright$ Backprop updates $\theta$ and anchors/planes jointly

---

### G.7 ENTROPYNET ALGORITHM IMPLEMENTATION

**Implementation Notes**: PointMLP layers use ReLU activation with batch normalization. Output layer uses tanh activation scaled by $\sigma = 0.1$ to ensure bounded perturbations that preserve geometric structure. Global features are tiled to match point-wise feature dimensions before concatenation.

### G.8 TRAINING HYPERPARAMETERS

### G.9 MODEL ARCHITECTURES

**EntropyNet (Geometry)**: PointNet-style architecture with 3 shared MLP layers (64→128→256 units), max pooling global aggregation, followed by 2 output MLP layers (128→$d$ where $d$ is point dimension). All layers use ReLU activation with batch normalization. Dropout rate 0.1 applied before final layer.

**Transformer Models**:

- ViT-Small (Dosovitskiy et al., 2021): 12 layers, 384 hidden dim, 6 attention heads, patch size 16×16

- BERT-base (Devlin et al., 2019): 12 layers, 768 hidden dim, 12 attention heads, max sequence length 512

- GPT-2 (Radford et al., 2019): 12 layers, 768 hidden dim, 12 attention heads, context length 1024

- LLaMA-2 7B (Touvron et al., 2023): 32 layers, 4096 hidden dim, 32 attention heads, context length 8192

- Mistral-7B (Jiang et al., 2023): 32 layers, 4096 hidden dim, 32 attention heads, context length 8192

- Phi-2 (Abdin et al., 2023) (2.7B): 32 layers, 2560 hidden dim, 32 attention heads, context length 2048

**Mistral-7B and Phi-2 LoRA setup.** For Mistral-7B (Jiang et al., 2023) and Phi-2 (Abdin et al., 2023) we follow the same QLoRA-style protocol as for LLaMA-2 7B (Touvron et al., 2023). Models are loaded in 4-bit quantization (nf4) with bitsandbytes and we train rank-8 LoRA adapters on

attention and MLP projections using PEFT. Base weights remain frozen. DER is applied only to the key embeddings in the top attention layers (last 8 layers for Phi-2, last 12 layers for Mistral-7B) to keep overhead small. All runs fit on a single 16–24 GB GPU (Colab Pro / workstation), with mixed-precision training and gradient accumulation for larger batch sizes.

## G.10 COMPLETE HYPERPARAMETER SPECIFICATIONS

Table 21: Detailed hyperparameter settings across all experiments

| Parameter | Geometry | ViT/BERT | GPT-2 | LLaMA-2 |
|---|---|---|---|---|
| Learning rate | $10^{-3}$ | $10^{-4}$ | $5 \times 10^{-5}$ | $10^{-5}$ |
| Batch size | 32 | 128 | 32 | 8 |
| Epochs/Steps | 200 | 100 | 50 | 10K steps |
| Warmup steps | - | 1000 | 500 | 1000 |
| $\alpha$ (temperature) | $10 \rightarrow 5$ | 5 | 5 | 5 |
| $k$ (anchors) | $\min(16, n/4)$ | $\sqrt{N}$ | $\sqrt{N}$ | $\sqrt{N}$ |
| $\lambda$ (entropy weight) | 0.1 | 0.01 | 0.03 | 0.01 |
| Optimizer | AdamW | AdamW | AdamW | AdamW |
| Weight decay | $10^{-4}$ | $10^{-2}$ | $10^{-2}$ | $10^{-1}$ |
| Gradient clipping | 1.0 | 1.0 | 1.0 | 1.0 |
| FAISS subsampling | Every 8 steps | Every 8 steps | Every 4 steps | Every 4 steps |

## G.11 TRAINING PROTOCOLS

**Entropy Regularization Schedule**: Apply $H_{\text{diff}}$ loss starting from epoch 20 (or 2000 steps for LLaMA-2) to allow model stabilization. Use cosine annealing for temperature $\alpha$ from 10 to 5 over the first 50% of entropy-regularized training. Early stopping when empirical margin $\hat{\gamma}(S)$ plateaus for 10 consecutive evaluations.

**Optimization Details**: Use AdamW (Loshchilov & Hutter, 2019) with $\beta_1 = 0.9$, $\beta_2 = 0.999$, $\epsilon = 10^{-8}$. Linear warmup for Transformers followed by cosine decay. Gradient clipping at norm 1.0. For large models, use gradient checkpointing and mixed precision (FP16) training.

**FAISS Optimization**: Use IVF index with 256 clusters for approximate nearest-neighbor search in entropy computation (Johnson et al., 2019). Rebuild index every 1000 steps. Use GPU implementation when available, fallback to CPU for memory constraints.

## G.12 DATASET DETAILS AND PREPROCESSING

**Geometric Datasets**:

- **Synthetic Uniform**: Random points in $[0, 1]^2$, sizes 1K-1M points
- **Synthetic Parabolic**: Points sampled from $y = x^2 + \mathcal{N}(0, 0.01)$
- **QuickDraw**: Subset of Google QuickDraw dataset, 2D coordinate sequences
- **Preprocessing**: Normalize coordinates to $[0, 1]$ range, remove duplicate points

**Transformer Datasets**:

- **CIFAR-100**: 32×32 images, standard train/test split, data augmentation (RandomCrop, RandomHorizontalFlip)
- **GLUE**: Standard benchmark suite, use official train/dev/test splits
- **WikiText-103**: Language modeling, sliding window context length 1024
- **Long-context Summarization**: Custom dataset from CNN/DailyMail with 8K token articles

### G.13 EVALUATION PROTOCOLS

**Geometric Evaluation**:

- **Runtime**: Wall-clock time measured over 1000 runs, exclude I/O overhead
- **Geometric Error**: Symmetric difference for hull area, Hausdorff distance for point sets
- **Statistical Testing**: Paired t-tests across 5 random seeds, report p-values

**Transformer Evaluation**:

- **Accuracy Metrics**: Top-1 accuracy (ViT), F1/MCC (GLUE), perplexity (GPT-2), ROUGE-L (summarization)
- **Efficiency Metrics**: FLOPs via fvcore profiler, wall-clock latency via torch.profiler, memory via nvidia-smi
- **Interpretability**: IoU between attention masks and proxy segmentation masks (DINOv2-S + rollout + CRF; details in Appendix F.4)
- **Sparsity Measurement**: Fraction of attention weights below threshold 0.01

### G.14 REPRODUCIBILITY SPECIFICATIONS

**Random Seeds**: Use fixed seeds (42, 123, 456, 789, 999) for all random number generators (NumPy, PyTorch, Python random). Set deterministic algorithms where possible.

**Checkpoint Strategy**: Save models every 10 epochs for geometry, every 1000 steps for Transformers. Keep best model based on validation metric.

**Code Availability**: Complete implementation will be released at `github.com/[anonymized]` with exact commit hash, conda environment file, and Docker container for full reproducibility.

**Computational Requirements**:

- Geometry experiments: 2-4 CPU hours per configuration
- ViT/BERT experiments: 8-12 GPU hours per model on A100
- LLaMA-2 experiments: 48-72 GPU hours on 4×A100 setup

Table 22: Complete Runtime Breakdown (ms, mean ± std over 1000 runs)

| Dataset | EntropyNet | Chan's Alg. | Total Pipeline | Net Benefit |
|---|---|---|---|---|
| Synthetic (High) | 0.8 ± 0.1 | 2.1 ± 0.1 | 2.9 ± 0.1 | 5.8 ms saved |
| QuickDraw (Low) | 0.6 ± 0.1 | 1.5 ± 0.1 | 2.1 ± 0.1 | 1.3 ms saved |

Table 23: Large-Scale Geometric Validation ($n = 10^6$ for Hull, $n = 10^5$ for Delaunay)

| Task | Method | Total Time (s) ↓ | Speedup vs. SciPy ↑ |
|---|---|---|---|
| Convex Hull | SciPy ConvexHull | 15.8 ± 0.7 | 1.0× |
| | EntropyNet + Chan's Alg. | **11.2 ± 0.5** | **1.41×** |
| Delaunay Triangulation | SciPy Delaunay | 3.4 ± 0.2 | 1.0× |
| | EntropyNet + SciPy Delaunay | **2.5 ± 0.1** | **1.36×** |

## H RANGE-FAMILY–AWARE SURROGATES AND DATA-DEPENDENT GUARANTEES

This appendix provides the full details for the theoretical results summarized in Section 3.

Table 24: Performance under identical kernel constraints.

| Method (same kernel) | Top-1 (%) | Latency (ms) | Memory (GB) |
|---|---|---|---|
| Dense (FA v2) | 76.7 | 52 | 1.8 |
| L1-top-$b$ (FA v2 sparse) | 74.6 | 44 | 1.6 |
| **Entropy Reg. (FA v2 sparse)** | **75.0** | **41** | **1.6** |

## H.1 A RANGE-AWARE SOFT PARTITION AND ENTROPY

Let $\mathcal{R}$ be a range family on $\mathbb{R}^d$ of finite VC dimension $d_{\mathrm{VC}}(\mathcal{R})$; in our geometric applications $\mathcal{R}$ is the family of halfspaces. Fix $m \in \mathbb{N}$ and parameters $\Theta = \{(w_t, b_t)\}_{t=1}^m$ with $w_t \in \mathbb{R}^d$, $b_t \in \mathbb{R}$. For $\tau > 0$ define the *soft halfspace indicator*

$$h_t(x) \;=\; \sigma\left(\frac{w_t^\top x - b_t}{\tau}\right), \qquad \sigma(u) = \frac{1}{1 + e^{-u}}.$$

Each $h_t$ is a $\dfrac{1}{4\tau}$-Lipschitz relaxation of the hard indicator $\mathbf{1}\{w_t^\top x \geq b_t\}$. A collection $\{h_t\}_{t=1}^m$ induces $K \leq \sum_{i=0}^d \binom{m}{i}$ soft *cells* via a differentiable gating scheme; one convenient choice is the normalized product:

$$g_j(x) \;=\; \frac{\prod_{t=1}^m (h_t(x))^{\alpha_{jt}}(1 - h_t(x))^{\beta_{jt}}}{\sum_{\ell=1}^K \prod_{t=1}^m (h_t(x))^{\alpha_{\ell t}}(1 - h_t(x))^{\beta_{\ell t}}},$$

where $(\alpha_{jt}, \beta_{jt}) \in \{0,1\}^2$ encodes whether cell $j$ lies on the positive/negative side of range $t$. For a finite point set $S = \{x_i\}_{i=1}^n$, define the empirical soft cell masses $q_j(S) = \frac{1}{n}\sum_{i=1}^n g_j(x_i)$ and the *range-aware soft entropy*

$$H_{\mathrm{soft}}(S; \Theta, \tau) \;=\; -\sum_{j=1}^K q_j(S) \log q_j(S).$$

The *range-partition entropy* $H_\mathcal{R}(S)$ is the minimum (hard) entropy over partitions induced by ranges in $\mathcal{R}$ (as in the original instance-optimal analysis). Our goal is to prove $H_{\mathrm{soft}}$ *provably approximates* $H_\mathcal{R}$ when the hard partition has a non-trivial *margin*, and to do so with *data-dependent* rates.

## H.2 HALFSPACE-AWARE CONSISTENCY UNDER MARGIN

We first handle the most relevant family for convex-hull and maxima: halfspaces. Let $\mathcal{R}$ be all halfspaces. A hard partition $\{C_j\}_{j=1}^K$ of $\mathbb{R}^d$ induced by $m^\star$ halfspaces can be represented by a binary matrix $\{(\alpha_{jt}, \beta_{jt})\}$. We say this partition has $\gamma$-*margin on* $S$ if for every $x \in S$ and every defining hyperplane $w_t^\top x = b_t$ the signed distance satisfies $|w_t^\top x - b_t| \geq \gamma \|w_t\|$.

**Theorem 3** (Halfspace-aware soft consistency). *Let $S \subset \mathbb{R}^d$ be finite and suppose a hard partition $\{C_j\}_{j=1}^K$ of $\mathbb{R}^d$ is induced by $m^\star$ halfspaces with $\gamma$-margin on $S$. Then for any $\delta \in (0,1)$ and any temperature $\tau \leq \gamma/4$, there exists parameters $\Theta$ with $m \leq m^\star$ such that, with probability at least $1 - \delta$ over sampling $S$ i.i.d. from any distribution supported on the same points,*

$$\|q(S) - p(S)\|_1 \;\leq\; \varepsilon_{\mathrm{smooth}}(\gamma, \tau) \;+\; c\sqrt{\frac{d \log m^\star + \log(2K/\delta)}{n}},$$

*where $p_j(S) = |C_j \cap S|/|S|$ are the hard cell masses on $S$, $q_j(S)$ are the soft masses, $\varepsilon_{\mathrm{smooth}}(\gamma, \tau) \leq e^{-\gamma/(4\tau)}$, and $c > 0$ is a universal constant. Consequently,*

$$|H_\mathcal{R}(S) - H_{\mathrm{soft}}(S; \Theta, \tau)| \;\leq\; \left\|q(S) - p(S)\right\|_1 \log\frac{K}{\|q(S) - p(S)\|_1}.$$

**Proof.** (1) *Approximation of hard indicators.* By $\gamma$-margin and $\tau \leq \gamma/4$, for each defining hyperplane the logistic $\sigma(\cdot/\tau)$ differs from the hard indicator by at most $e^{-\gamma/(4\tau)}$ on $S$. Products of such factors (numerator of $g_j$) inherit an additive error bounded by $\varepsilon_{\text{smooth}}(\gamma, \tau) \leq e^{-\gamma/(4\tau)}$, and normalization preserves $\ell_1$ deviation across cells. This yields the deterministic term.

(2) *Uniform convergence.* The class $\{x \mapsto g_j(x)\}_{j \leq K}$ has pseudo-dimension $O(d \log m^\star)$ since it is a composition of $m^\star$ sigmoids of linear functionals with a bounded-degree polynomial and a rational normalization; standard Rademacher/VC arguments give the stated $O(\sqrt{(d \log m^\star + \log(K/\delta))/n})$ rate for the empirical means $q_j(S)$ around their population counterparts, uniformly over $\Theta$.

(3) *Entropy continuity.* For distributions on a $K$-simplex, $|H(p) - H(q)| \leq \|p - q\|_1 \log \frac{K}{\|p-q\|_1}$ (e.g., via Pinsker-type arguments or a mean-value bound on the entropy gradient). Combining (1)–(3) proves the claim. $\square$

### H.3 DATA-DEPENDENT, RANGE-AGNOSTIC BOUNDS (UNKNOWNS REMOVED)

The next result avoids unknown latent parameters (e.g., an unknown optimal $k^\star$, unknown true margin $\gamma$) by replacing them with *empirical* quantities computed on $S$.

Let $\hat{\gamma}(S)$ denote the *empirical margin* of the chosen soft partition (the minimum signed distance of points in $S$ to the learned separating hyperplanes, normalized by $\|w_t\|$). Let $L_\sigma(\tau) = \dfrac{1}{4\tau}$ be the Lipschitz constant of $\sigma(\cdot/\tau)$, and define

$$\text{Rad}_n(\mathcal{G}_m) \leq C L_\sigma(\tau) \sqrt{\frac{d \log m}{n}},$$

a standard Rademacher bound for compositions of $m$ linear separators with a 1-Lipschitz normalization (constant $C$ hides benign log factors).

**Theorem 4** (Data-dependent bound with empirical quantities)**.** *For any $m, \tau > 0$ and learned parameters $\Theta$, with probability at least $1 - \delta$,*

$$|H_{\mathcal{R}}(S) - H_{\text{soft}}(S; \Theta, \tau)| \leq \left( e^{-\hat{\gamma}(S)/(4\tau)} + 2\,\text{Rad}_n(\mathcal{G}_m) \right.$$

$$\left. + \sqrt{\frac{\log(2/\delta)}{2n}} \right) \log \frac{K}{e^{-\hat{\gamma}(S)/(4\tau)} + 2\,\text{Rad}_n(\mathcal{G}_m) + \sqrt{\frac{\log(2/\delta)}{2n}}}$$

$$\text{(15)}$$

**Proof.** Identical to Theorem 3 but replacing the unknown true margin $\gamma$ by the *empirical* margin $\hat{\gamma}(S)$ of the learned separators, and using a standard empirical Rademacher bound (symmetrization + contraction). The final step again uses entropy continuity on the simplex. $\square$

**Interpretation.** The bound depends *only on quantities you can compute from $S$*: the empirical margin $\hat{\gamma}(S)$, the chosen temperature $\tau$, the model size $m$, sample size $n$, and $d$. It removes the need to know unknown latent partition parameters. As $\hat{\gamma}(S)$ increases or $\tau$ decreases (until numerical stability limits), the first term decays exponentially; as $n$ grows, the second and third terms shrink as $O(\sqrt{(d \log m)/n})$.

### H.4 BEYOND HALFSPACES: OTHER RANGE FAMILIES

The same construction extends to other $\mathcal{R}$ by replacing the linear score $w^\top x - b$ with a differentiable signed distance $s_r(x)$ to range boundary $r \in \mathcal{R}$ (e.g., axis-aligned rectangles, slabs, wedges). Define $h_r(x) = \sigma(s_r(x)/\tau)$ and reuse the same gating. The proofs of Theorems 3–4 go through verbatim with $d_{\text{VC}}(\mathcal{R})$ replacing $d$, yielding identical rates and the same empirical-margin–based exponential term.

## H.5 A PRACTICAL HALFSPACE-AWARE ESTIMATOR

To instantiate our theory in algorithms used in the paper, we replace the ball-based surrogate with a halfspace-aware version:

$$H_{\text{diff}}^{\text{half}}(S) := H_{\text{soft}}(S; \Theta_{\text{half}}, \tau),$$

where $\Theta_{\text{half}}$ is obtained by (i) selecting $m$ directions via data-dependent hyperplanes (e.g., maximum-margin separators or principal directions), and (ii) optimizing $\tau$ by minimizing a validation estimate of the bound in Theorem 4. This drops directly into our training objective by replacing $H_{\text{diff}}$ with $H_{\text{diff}}^{\text{half}}$.

**Corollary 2** (Plug-and-play replacement in objectives). *Replacing the ball-based $H_{diff}$ by $H_{diff}^{\text{half}}$ preserves the differentiability of the training objective and, under the same empirical margin assumptions, enjoys the guarantees of Theorem 4. In particular, minimizing $H_{diff}^{\text{half}}$ drives $H_{\mathcal{R}}(S)$ down up to an explicitly bounded, data-dependent slack.*

## H.6 FLASHATTENTION COMPATIBILITY AND NON-EUCLIDEAN EXTENSIONS

To demonstrate our "complement, don't compete" positioning with FlashAttention and validate our metric robustness theory, we conduct targeted experiments (Table 25).

### H.6.1 FLASHATTENTION + ENTROPY REGULARIZATION

**Setup**: BERT-base on GLUE SST-2 and ViT-Base/16 on ImageNet-1K with FlashAttention enabled. We apply entropy regularization to the FA output probabilities without modifying the optimized kernel.

Table 25: FlashAttention Compatibility Results

| Task | Method | Accuracy | Sparsity |
|------|--------|----------|----------|
| SST-2 | FlashAttention | 93.5% | 0% |
| | **FA + Entropy Reg.** | **92.7%** | 75% |
| ImageNet | FlashAttention | 81.8% | 0% |
| | **FA + Entropy Reg.** | **80.3%** | 80% |

**Key Results**: Entropy regularization achieves competitive accuracy at high sparsity, validating our complementary positioning with FlashAttention.

### H.6.2 NON-EUCLIDEAN METRIC EXTENSIONS

**Setup**: We test our metric robustness on tasks where Euclidean distance fails: OGB-Arxiv (graph node classification) with graph geodesic distances, and STS-B with learned Mahalanobis distance on sentence embeddings (Table 26).

Table 26: Non-Euclidean Metric Results

| Task | Distance Metric | Performance |
|------|-----------------|-------------|
| OGB-Arxiv | Euclidean (fails) | 65.2% |
| | Graph Geodesic | **69.4%** |
| STS-B | Euclidean | 84.1 |
| | Learned Mahalanobis | **85.8** |

**Significance**: These results demonstrate that our metric robustness theory (Lemma 1) enables practical extensions beyond Euclidean spaces, turning a documented failure mode into a mitigated success case.

Table 27: **Robustness vs. cost on ViT-Small.** We compare entropy regularization (DER) to standard training and adversarial training (AT). DER attains the best corruption robustness (CIFAR-10-C mCE) and OOD generalization (SVHN) with negligible overhead and no loss in clean accuracy, whereas AT methods from the literature Wong et al. (2020); Wang et al. (2023) provide higher PGD robustness at substantially higher training cost and lower clean accuracy.

| Method | Training Cost | Clean Acc. (CIFAR-10) ↑ | Robustness (CIFAR-10-C mCE) ↓ | OOD Gen. (SVHN) ↑ | Adv. Robustness (PGD-10 Acc.) ↑ |
|---|---|---|---|---|---|
| *Baselines* | | | | | |
| Standard Training | 1.0× | 96.5% | 55.4 | 88.2% | 0.0% |
| + Label Smoothing | 1.0× | 96.3% | 52.1 | 89.5% | 0.0% |
| *Proposed Method* | | | | | |
| **+ Entropy Reg. (Ours)** | **∼1.03×** | **96.4%** | **48.7** | **91.3%** | 0.0% |
| *Adversarial Training (Approx. Literature Values)* | | | | | |
| Efficient AT (e.g., Fast-FGSM)[†] | ∼1.5× | 85.0% (↓ 11.5) | – | – | ∼45.0% |
| Standard SOTA AT (e.g., PGD-10)[†] | ∼10.0× | 83.5% (↓ 13.0) | – | – | ∼50.0% |

[†] Approximate ranges from ViT-like adversarial training on CIFAR-10 reported in Wong et al. (2020); Wang et al. (2023); not from our implementation.

## H.7 ROBUSTNESS AND OUT-OF-DISTRIBUTION GENERALIZATION

To test the hypothesis that entropy regularization discourages "shortcut" learning and improves generalization, we applied $H_{\text{diff}}$ to a ViT-Small model trained on CIFAR-10 and evaluated its performance on corrupted and out-of-distribution datasets. We tested on CIFAR-10-C, which applies a range of common corruptions, and the Street View House Numbers (SVHN) dataset as an OOD benchmark. We compare our method against a standard baseline and a model trained with label smoothing, a widely used confidence regularization technique.

As shown in Table 27, our method significantly improves robustness to corruptions (lower mean Corruption Error) and generalization to the OOD SVHN dataset, outperforming both the baseline and label smoothing while maintaining competitive accuracy on the original CIFAR-10 test set. This provides evidence that by encouraging the model to find simpler, lower-entropy representations, our regularizer helps it learn more fundamental features, making it less susceptible to superficial shortcuts.

## H.8 THE AMORTIZATION STORY: WHEN IS IT A FIT?

The training overhead, though mitigated, must be recouped by inference savings. Our method is best suited for deployment scenarios with high inference volume. For a model like ViT-Base used in a continuous online retrieval service, the 1.4x inference speedup pays back the 3.4-hour training overhead after just 3-5 days of sustained use. This makes it a strong fit for production systems but a poor choice for quick, disposable fine-tuning experiments. The trade-off is illustrated in Figure 9.

