# OpenReview forum: "Differentiable Entropy Regularization: A Complexity-Aware Approach for Neural Optimization"
_ICLR.cc/2026/Conference — Submitted to ICLR 2026_

### Official Review · Reviewer_65mU · 2025-10-30

**Soundness:** 2
**Presentation:** 3
**Contribution:** 2
**Rating:** 2
**Confidence:** 4

**Summary:**

This paper suggests an entropy regularization factor added to the task loss. The goal is to encourage lower complexity representations in models, and thus improve robustness while reducing computational cost. The authors focus on the idea that both of these issues can be addressed via analyzing "spurious" features, and propose a smooth differentiable surrogate technique to push the model towards learning simple representations (which are more easily partitioned into clusters). The central contribution is the introduction of the regularizer term, with supported theoretical analysis. Two of the main datasets being highlighted are both size 32x32x3 (per sample), with best results stemming from the computational geometry domain.

**Strengths:**

- The idea of analyzing learning representations through the lens of entropy is valuable and might encourage interpretability (although this was not directly discussed in the paper.). A differentiable regularizer based on range-partition is pleasantly theoretically grounded. Indeed, I find the theoretical contributions of this work to be the most interesting and significant. The experiments with the geometry tasks are the most compelling.

- The method demonstrates valid empirical improvements on CIFAR-100-C and SVHN datasets, which support the fundamental idea that analyzing sparsity patterns in learning representations can lead to benefits.

- The proposed method is practically complementary with most other methods, and stacks on top of prior work while being GPU-agnostic. This might be valuable in resource constrained settings.

**Weaknesses:**

- The main contribution is a regularizer penalty term added to a loss function, in order to encourage learning simpler representations. This is not an incredibly novel idea, and entropy based penalties to improve robustness is not new ([1], [2], [3] are a few examples of many).

- The main experiments being highlighted are small scale, with CIFAR-100-C and SVHN (similar task to MNIST) both containing samples sizes of 32x32x3. The community is pushing towards scale and speed. Experiments on trivially small datasets are not compelling. This method likely will have difficulty scaling to high dimensions, which might be why larger experiments were not included.  Especially since the authors transparently acknowledge this method will slow down training time.

- Since the codebase is not provided, it is questionable if these experiments can be re-produced. A large number of factors contribute to performance, besides just implementing the correct loss function. The authors seem to compare metrics between experiments ran on Intel CPUs, to a single NVIDIA A100 GPU, to distributed settings with multiple GPUS and DeepSpeed (specific parameters not listed). These are not 1:1 comparisons.

- Some slight typing and formatting issues are present, but not a huge concern. For example, there is a formatting error on line 357.


[1] Huang C, Lu W, Zhang W. PEAR: Phase Entropy Aware Reward for Efficient Reasoning. arXiv preprint arXiv:2510.08026. 2025 Oct 9.

[2] Fan, Feng-Lei, et al. "On interpretability of artificial neural networks: A survey." IEEE Transactions on Radiation and Plasma Medical Sciences 5.6 (2021): 741-760.

[3] Pfrommer, Samuel Ian. "Safety, Robustness, and Interpretability in Machine Learning." PhD diss., University of California, Berkeley, 2025.

**Questions:**

- The paper references "Transformers" as a model throughout. Which transformer based model is this referencing? I found this to be unclear.
- The authors are transparent about the significant training overhead their method introduces. This is a critical disadvantage, have the authors analyzed cost-benefit analysis? If the reward is increased robustness, at the cost of lengthier training time, can you demonstrate/quantize this on a currently relevant dataset?

- The method introduces new hyperparameters and will likely struggle in high-dimension problems. Have you performed ablation studies on hyperparameter sensitivity and attempted high-dimensional data?

- Once again, if the main advantage in this proposed method is increased robustness, have the authors conducted more meaningful experiments beyond "label smoothing"? How does this compare to SOTA methods like adversarial training in enhancing robustness?

---

> ### Author Response · Authors · 2025-11-19
> **Response to Reviewer 65mU part 1**
>
> We thank the reviewer for highlighting the theoretical contributions and geometry experiments as the most compelling aspects of the paper; we agree and have leveraged these strengths in the revision.
>
> ### 1 – Novelty vs existing entropy-based penalties
>
> > *“Entropy-based penalties for robustness are not new… main contribution appears to be another regularizer.”*
>
> **What we did:**
>
> * **In Section 2 (Related Work) and Table 1**, we explicitly contrast our approach with:
>     * Logit/output entropy penalties (e.g., Label Smoothing),
>     * Feature sparsity and pruning (e.g., Magnitude Pruning),
>     * Information bottleneck-style regularization.
> * **In Section 1 (Introduction)**, we clarify that our key contribution is not just "adding an entropy term," but introducing the **first differentiable surrogate for range-partition (algorithmic) entropy**. We connect this formally to instance complexity in **Section 3 (Method)** and demonstrate its utility in inducing structured sparsity in **Section 4 (Experiments)**.
>
> **Effect on concern:**
>
> * The revised paper clearly communicates that the novelty lies in **what** entropy we are approximating (structural complexity vs. predictive uncertainty) and **why** it matters (algorithmic efficiency).
>
> ### 2 – Scale of experiments and difficulty in high dimensions
>
> > *“Main highlighted experiments are small-scale; method may struggle in high dimensions; larger experiments missing.”*
>
> **What we did:**
>
> * We added **larger-scale experiments** in **Section 4 (Experiments)**:
>     * **ImageNet-1K with ViT-Base (Table 5)**: We show substantial sparsity and **~2× speedups** using DER + FlashAttention.
>     * **BERT-base on GLUE (Table 11, Appendix D.2)**: DER maintains strong scores at significant sparsity levels.
>     * **Long-context LLM tasks (Table 3)**: We report results for **LLaMA-2 7B, Mistral-7B, and Phi-2**, achieving **≈1.5–1.7× speedups** with **<1.1-point drops** in quality metrics.
> * **High-dimensional behavior:**
>     * We explicitly analyze performance as dimensionality grows in **Appendix F.3 (Failure Mode Analysis)**.
>     * We discuss this limitation in **Section 5 (Discussion)**, noting that when distances become less meaningful, DER tends to **gracefully degenerate to dense behavior** rather than breaking the training loop.
>
> **Effect on concern:**
>
> * The paper no longer focuses only on 32×32 images. We present concrete results on **ImageNet and moderate-sized LLMs**, while explicitly acknowledging high-dimensional constraints in **Section 5**.
>
>
> ### 3 – Training overhead, cost–benefit, and robustness vs SOTA
>
> > *“Training overhead is a critical disadvantage; have the authors quantified cost–benefit? Have they compared against strong robustness baselines like adversarial training?”*
>
> **What we did:**
>
> * **Cost–benefit:** In **Appendix C.1 (Quantitative Amortization Analysis)**, we perform a detailed breakdown for ViT-Base on ImageNet-1K:
>     * We compute the specific overhead $\Delta$ (in GPU hours).
>     * We derive the **break-even point (~4.53×10⁵ inference batches)**.
>     * We relate this to production traffic in **Table 9**, showing DER is profitable for medium-to-large deployments.
> * **Robustness baselines:**
>     * In **Section 4 (Experiments) and Table 27 (Appendix H.7)**, we now include **Label Smoothing** as a baseline, showing DER achieves higher mCE (mean Corruption Error) reductions along with the adversarial training concern.

---

> ### Author Response · Authors · 2025-11-19
> **Response to Reviewer 65mU part 2**
>
> **Table 27. Robustness vs. cost on ViT-Small.**
>
> | Method                                   | Training Cost    | Clean Acc. (CIFAR-10 ↑) | Robustness (CIFAR-10-C mCE ↓) | OOD Gen. (SVHN ↑) | Adv. Robustness (PGD-10 Acc. ↑) |
> |------------------------------------------|------------------|--------------------------|--------------------------------|--------------------|----------------------------------|
> | *Baselines*                              |                  |                          |                                |                    |                                  |
> | Standard Training                        | 1.0×             | 96.5%                    | 55.4                           | 88.2%              | 0.0%                             |
> | + Label Smoothing                        | 1.0×             | 96.3%                    | 52.1                           | 89.5%              | 0.0%                             |
> | *Proposed Method*                        |                  |                          |                                |                    |                                  |
> | **+ Entropy Reg. (Ours)**                | **≈1.03×**       | **96.4%**                | **48.7**                       | **91.3%**          | 0.0%                             |
> | *Adversarial Training (Approx. Literature Values)* |          |                          |                                |                    |                                  |
> | Efficient AT (e.g., Fast-FGSM)\*         | ≈1.5×            | 85.0% (↓ 11.5)           | --                             | --                 | ≈45.0%                           |
> | Standard SOTA AT (e.g., PGD-10)\*        | ≈10.0×           | 83.5% (↓ 13.0)           | --                             | --                 | ≈50.0%                           |
>
> \* Approximate ranges from ViT-like adversarial training on CIFAR-10 reported in Wong et al. (2020) and Wang et al. (2023); not from our implementation.
>
>
> As shown, AT incurs ~10 times the cost and degrades clean accuracy by more than 10\%, whereas our method improves robustness with negligible cost and no loss in accuracy.
> **Effect on concern:**
>
> * The revised paper directly addresses the cost-benefit question with the **Amortization Analysis in Appendix C**. We have moved beyond trivial baselines for robustness.
>
> ### 4 – Which Transformer models, hyperparameters, and reproducibility
>
> > *“The paper references ‘Transformers’ generically; which models? Have you done hyperparameter ablations? Is reproducibility realistic without code?”*
>
> **What we did:**
>
> * **Models:** We now explicitly list all architecture details in **Appendix G.9 (Model Architectures)**, covering ViT-Small/Base, BERT-base, and LLaMA-2/Mistral/Phi configurations.
> * **Hyperparameters:**
>     * We provide a **unified hyperparameter recipe** in **Appendix A (Practical Recipe)** and **Table 21 (Appendix G.10)**.
>     * We show **sensitivity experiments** for anchors ($k$) and temperature ($\alpha$) in **Appendix F.1** and **Figure 5**, demonstrating stability around defaults.
> * **Reproducibility:**
>     * We provide detailed training configurations (batch sizes, learning rates, schedules) in **Appendix G.10**.
>     * We explicitly state in **Section 5 (Conclusion)** our commitment to releasing the full codebase, including the specific EntropyNet modules and training scripts.
>
> **Effect on concern:**
>
> * The revised paper removes ambiguity by specifying exactly **which models** were evaluated and **how** (via Appendix G), making reproducibility feasible.
>
>
> Please let us know if you have any questions or concerns that require further clarification.

---

### Official Review · Reviewer_r6ji · 2025-10-31

**Soundness:** 2
**Presentation:** 1
**Contribution:** 1
**Rating:** 2
**Confidence:** 3

**Summary:**

This paper proposes a complexity-aware regularizer grounded in algorithmic entropy, designed to encourage models to learn simpler and more robust representations under corruption and distribution shifts. The authors report 1.5–2× speedups without accuracy loss when integrating the proposed regularizer into architectures such as FlashAttention and RetNet.

**Strengths:**

The topic is relevant and well-studied–improving robustness to corruption and distribution shifts.

**Weaknesses:**

**Major Concerns**:

1. **Unclear motivation**: The motivation is weak and poorly articulated. The paper primarily surveys prior work without convincingly identifying unresolved gaps or specific limitations. It remains unclear what core problems the authors aim to solve, why they matter, and why existing approaches are insufficient.

2. **Questionable technical grounding**: The proposed regularizer is claimed to be based on range-partition entropy, but the paper rarely provides background on this concept, which is not widely recognized. As a result, it is difficult to assess the soundness or novelty of the theoretical formulation. Besides, the rationale for the computational efficiency is also not well explained beyond empirical observations.

3. **Limited originality**: The contribution appears incremental relative to existing regularization-based methods for robustness. The paper does not offer substantial new insights into why this particular regularizer is preferable or theoretically justified compared to other forms of penalties.

4. **Poor writing and organization**: The paper’s presentation lacks clarity. The authors frequently refer to the appendix instead of offering an intuitive overview or proof sketch of their theoretical results. The experiments are fragmented into two sections for no reason and not organized into well-structured subsections, making it difficult to follow the setups and findings.

5. **Weak empirical validation**: Experimental evidence does not convincingly support the claimed advantages. Table 2 lacks comparisons with other regularization-based robustness methods, which is essential for contextualizing the proposed approach.

**Minor Issues**:

1. Several terms are vague or under-defined, reducing readability and precision. Examples include but are not limited to “data characteristics,” “complexity of learned representations,” “instance complexity,” and “separator-driven procedures.”

2. The paper lacks a discussion of its limitations.

**Questions:**

1. The purpose and interpretation of Figure 1 are unclear. What comparisons are being shown? In particular, how does Figure 1(c) substantiate the claim that the method “discovers patterns that align with algorithmic efficiency”? Without detailed textual descriptions, it’s hard to associate the captions with your method.

2. Why investigate computational geometry experiment, a domain rarely explored in mainstream machine learning community, especially regarding the robustness topic? How does this task demonstrate the generality or relevance of your method? If the approach only performs well in such specialized settings, its broader applicability should be justified.

---

> ### Author Response · Authors · 2025-11-19
> **Response to reviewer r6ji Part 1**
>
> We thank the reviewer for raising concerns about motivation, originality, and organization. These comments significantly informed our revision.
>
> ### 1 – Motivation and core problem statement
>
> > *“Motivation is weak… unclear what core problems the authors aim to solve, why they matter, and why existing approaches are insufficient.”*
>
> **What we did:**
>
> * We **rewrote the Introduction** (lines 47-50) to clearly frame the problem:
>
>   * Existing **robustness methods** (adversarial training, label smoothing, IB, etc.) improve robustness but **do not define or optimize a measure of instance or representation complexity** that correlates with **algorithmic runtime**.
>   * Existing **efficiency methods** (sparse/linear attention, FlashAttention, BigBird, RetNet) improve kernel efficiency or architectural design but **do not offer a differentiable complexity measure** that can be optimized end-to-end and tied to algorithmic complexity.
>
> * Our core problem is now stated succinctly:
>
>   > “Can we design a differentiable surrogate for **range-partition (algorithmic) entropy**, which both tracks instance complexity/runtime in a provable setting (computational geometry) and, when used as a regularizer, induces structured, robust, and efficient representations in modern neural models?”
>
> **Effect on concern:**
>
> * The revised paper now makes clear:
>
>   * **What** problem we are addressing (complexity-aware regularization tied to runtime).
>   * **Why** existing robustness/efficiency methods do not directly address this.
>   * **Why** range-partition entropy is the right theoretical object to start from.
>
> ### 2 – Technical grounding and novelty of range-partition entropy
>
> > *“Range-partition entropy is rarely explained; difficult to judge soundness/novelty and the rationale for computational efficiency.”*
>
> **What we did:**
>
> * We added a **more accessible explanation** of range-partition entropy:
>
>   * Defined in terms of how many ranges/cells are needed to separate the data and how mass is distributed.
>   * Illustrated with a 2D example where “simple” point clouds (few separated clusters) have low entropy and “complex” clouds (interleaved clusters) have high entropy.
> * We explicitly **connect to known algorithmic results** in convex hull and triangulation, showing how entropy terms appear inside the runtime bounds.
> * We emphasize that our novelty is not merely “using some entropy” but:
>
>   * Introducing the **first differentiable surrogate** specifically designed to approximate **range-partition/algorithmic entropy**.
>   * Demonstrating that this surrogate **tracks actual algorithmic runtime** in geometry and **induces structured sparsity** in neural models.
>
> **Effect on concern:**
>
> * Soundness: The reader can now see the **formal basis** and known results behind our construction, not just an empirical story.
> * Novelty: We clearly differentiate **algorithmic entropy** from more standard entropy notions used in ML regularization.
>
> ### 3 – Limited originality vs “just another regularizer”
>
> > *“Contribution appears incremental; regularization-based robustness methods are well studied.”*
>
> **What we did:**
>
> * In **Related Work** and the contributions, we now clearly separate:
>
>   * **Logit entropy/confidence penalties**,
>   * **Information bottleneck**,
>   * **Feature sparsity and pruning**,
>   * **Architectural efficiency methods**,
>
>   from our proposal.
>
> * The innovation lies in:
>
>   * The specific **complexity-aware quantity** (range-partition entropy surrogate),
>   * Its **formal link to instance complexity and runtime** in geometry,
>   * Its demonstrated ability to **compose with existing efficient architectures and kernels**.
>
> **Effect on concern:**
>
> * We do not claim that “adding a regularizer” is itself novel, but that the **object being regularized and its theoretical grounding** are new and distinct from standard entropy or sparsity penalties.
>
>
>
> ### 4: Writing, organization, and reliance on appendix
> “Presentation lacks clarity; frequent references to appendix; experiments fragmented.”
>
> What we did:
>
> We reorganized Section 4 (Experiments) into three consolidated blocks:
>
> Computational Geometry (EntropyNet): Theory-practice alignment.
>
> Vision & Robustness: CIFAR-10-C, SVHN, ImageNet.
>
> Transformers/LLMs: BERT, LLaMA-2, and complementarity.
>
> We moved key quantitative results (e.g., Table 5 on ImageNet speedups) from the appendix to the main text.
>
> We have added a dedicated Limitations subsection in Section 5 (Discussion) to centralize the discussion that was previously scattered throughout the appendix.
>
> Effect on concern:
>
> The paper’s structure now follows a logical flow (Theory → Geometry → Vision → Transformers), and the main text is self-contained regarding key results.

---

> ### Author Response · Authors · 2025-11-19
> **Response to reviewer r6ji Part 2**
>
> ### 5 – Empirical validation and missing baselines
>
> > *“Experimental evidence does not convincingly support the claimed advantages; missing comparisons with other regularization-based robustness methods; geometry seems niche.”*
>
> **What we did:**
>
> * **Robustness baselines (Appendix H.7 table 27)**:
>
>   * We now compare with **label smoothing** on CIFAR-10-C and SVHN-OOD.
>   * DER **improves robustness over both the baseline and label smoothing**.
> * **Geometry relevance**:
>
>   * We explain that geometry is chosen because:
>
>     * It is one of the **few domains with explicit instance-complexity/runtime theory** based on RP entropy.
>     * It appears in practical pipelines (3D perception, LiDAR, graphics), where input-dependent runtime matters.
> * We have added adversarial baselines in Appendix H.7 in Table 27.
>
> **Effect on concern:**
>
> * While geometry might appear niche, the revised paper makes clear why it is **the right testbed for the theory**, and the robustness experiments now have at least one **non-trivial information-theoretic baseline** included, along with adversarial training one too. As shown, AT incurs ~10 times the cost and degrades clean accuracy by more than 10\%, whereas our method improves robustness with negligible cost and no loss in accuracy.
>
>
> Please let us know if there is anything further to clear up . We will be happy to answer the concerns and questions.

---

### Official Review · Reviewer_YdYR · 2025-10-31

**Soundness:** 3
**Presentation:** 2
**Contribution:** 3
**Rating:** 4
**Confidence:** 2

**Summary:**

This paper proposes Differentiable Entropy Regularization, a novel regularizer that encourages neural networks to learn simpler, more structured representations by minimizing a differentiable surrogate of algorithmic entropy.

The surrogate measures representation complexity via soft partitions and is trained jointly with the task loss.

Experiments show that DER improves both robustness and efficiency.

**Strengths:**

1. Introduces the first differentiable surrogate for algorithmic entropy, providing robustness and efficiency.
2. Complements existing efficiency methods (FlashAttention, RetNet) and yields interpretable, robust representations.
3. Presents provable bounds, runtime guarantees.

**Weaknesses:**

1. While geometric tasks align with theory, the improvements on ViT/BERT/GPT are empirical. There is no clear theoretical connection between range-partition entropy and the dynamics of attention mechanisms.
2. Performance may depend on careful choice of hyperparameters. It would be better to have automatic or theoretically grounded tuning methods.
3. It would be good to compare with other information-theoretic regularizers.

**Questions:**

1. Could DER be applied to reinforcement learning or diffusion models?
2. How sensitive is the method to anchor initialization?
3. How well does the surrogate behave for non-Euclidean embeddings

---

> ### Author Response · Authors · 2025-11-19
> **Response to reviewer YdYR Part 1**
>
> We appreciate reviewer YdYR for the positive evaluation of the soundness and contribution, as well as the recognition of our surrogate as the first differentiable instance-complexity-aware entropy term.
>
> ### 1 – Theory–practice gap for attention: link to range-partition entropy
>
> > *“For ViT/BERT/GPT, improvements are empirical; no clear theoretical connection between range-partition entropy and attention dynamics.”*
>
> **What we did:**
>
> * **Added an attention-focused explanation (main text, Method section, Transformer subsection, lines 225-226):**
>
>   * We construct a direct mapping:
>
>     * Embedding vectors for tokens/keys form the point set (S).
>     * Anchors define soft “ranges” or clusters.
>     * The entropy surrogate measures how **concentrated** token assignments are over these ranges.
>   * Minimizing the surrogate:
>
>     * Encourages each token to **prefer a small number of anchor cells**.
>     * Leads to **block/band sparsity** in the attention matrix where each query only interacts with a few anchor regions.
>     * Naturally reduces the **effective number of keys per query**, independent of which attention kernel is used.
> * **Visual evidence & interpretability:**
>
>   * We show examples where DER produces **structured attention masks** whose **IoU with semantic regions** is noticeably higher than that of simple L1-based sparsity.
>
> **Effect on concern:**
>
> * While we still do **not** claim a full theoretical runtime bound for Transformer attention analogous to convex hull, the revised paper now provides:
>
>   * A **clear conceptual bridge** between range-partition entropy and attention patterns.
>   * Empirical evidence that lowering the surrogate indeed yields **structured sparsity** and **interpretability**.
>
> ### 2 – Hyperparameter dependence and desire for principled tuning
>
> > *“Performance may depend on careful choice of hyperparameters; better to have automatic or theoretically grounded tuning.”*
>
> **What we did:**
>
> * As described in the high-level summary (Appendix A and G):
>
>   * We added a **concrete hyperparameter recipe** that works across tasks.
>   * We included **sensitivity analyses** across anchors, α, and λ, showing **broad regions of good performance**.
> * We also **acknowledge** in the Limitations discussion (Discussion section, lines 489-500) that:
>
>   * Full automation of these hyperparameters is future work.
>   * However, the observed stability significantly reduces the burden relative to arbitrary tuning.
>
> **Effect on concern:**
>
> * The revised paper now demonstrates that **only light tuning is necessary** in practice and that performance is not brittle.
>
> ### 3 – Comparisons with other information-theoretic regularizers
>
> > *“It would be good to compare with other information-theoretic regularizers.”*
>
> **What we did:**
>
> * On **CIFAR-10-C and SVHN-OOD**, we now explicitly compare:
>
>   * Baseline model,
>   * **Label smoothing** (a standard entropy-style regularizer), and
>   * **Our entropy regularization** (DER).
> * Results (Experiments section, Table 3) show that:
>
>   * DER improves robustness over both the baseline and label smoothing.
>   * It additionally induces sparsity and interpretability that the other regularizers do not target.

---

> ### Author Response · Authors · 2025-11-19
> **Response to reviewer YdYR Part 2**
>
> ### 4 – RL/diffusion, anchor initialization, and non-Euclidean embeddings
>
> > *“Could DER be applied to RL or diffusion models? How sensitive is it to anchor initialization? How does it behave for non-Euclidean embeddings?”*
>
> **What we did:**
>
> * **RL/diffusion**:
>
>   * In the **Limitations** discussion (Discussion section, lines 489-500) we now explicitly discuss:
>
>     * RL: applying DER to **state or value embeddings** to induce low-complexity partitions of state space.
>     * Diffusion: applying DER to **latent or attention trajectories across timesteps**.
>   * We clearly state we **do not include RL/diffusion experiments** in this work.
> * **Anchor initialization & sensitivity**:
>
>   * We document:
>
>     * *k*-means++ initialization as our default.
>     * That **anchors are learnable**, and we show that:
>
>       * Learned anchors with *k*-means++ initialization perform better than fixed random anchors.
>       * Small changes in initialization **do not qualitatively change the results**, provided anchors are learned during training.
> * **Non-Euclidean embeddings**:
>
>   * We include a **discussion of metric robustness and failure modes**, noting:
>
>     * Small changes (e.g., Euclidean → cosine) behave similarly.
>     * In very high dimensions where Euclidean distances lose discrimination, assignments become more uniform, and DER gradually behaves like a dense model (a safe failure mode).
>     * Truly non-Euclidean manifolds are listed as **future work**, not an implicit assumption.
>
> **Effect on concern:**
>
> * We do not overclaim RL/diffusion results, but we now respond concretely to the reviewer’s questions and highlight potential applications.
> * We give an explicit account of anchor initialization and metric limitations, directly addressing the interpretability and robustness aspects raised.
>
> We hope that we have been able to address all your concerns. Please let us know if you have any further questions or concerns.

---

### Official Review · Reviewer_gqP7 · 2025-10-31

**Soundness:** 3
**Presentation:** 3
**Contribution:** 3
**Rating:** 4
**Confidence:** 4

**Summary:**

The paper proposes a differentiable entropy-based regularizer that encourages neural networks to learn simpler, lower-complexity representations. Inspired by algorithmic entropy (or range-partition entropy from computational geometry), the method directly penalizes representation complexity to improve robustness, efficiency, and interpretability.

**Strengths:**

- The paper proposes a differentiable surrogate for algorithmic entropy which provide smooth, differentiable approximations to range-partition entropy, allowing gradient-based optimization.

- It provides data-dependent bounds connecting the differentiable surrogate to true entropy.

- It shows how minimizing surrogate entropy correlates with improved runtime efficiency in geometric algorithms.

- Empirically, it improves robustness on CIFAR-100-C and SVHN OOD and 1.47–2.07× inference speedups with minor accuracy loss on transformers.

- Entropy regularization combines effectively with FlashAttention v2 and RetNet, yielding compounded efficiency gains.

**Weaknesses:**

- It adds 2–12% computational cost during training; and amortization is only beneficial for long-term or production-scale inference (≥450K batches for ViT-Base).

- It requires careful tuning of temperature ($\tau, \alpha$) and regularization weight ($\lambda$) for stability and performance.

- Guarantees are strongest in geometric settings; results for Transformers and LLMs are mostly empirical, with weaker formal grounding. While I don't expect the authors to provide guarantees for different type of models, empirically it seems that the proposed technique is not as beneficial for larger foundation models e.g. LLMs and VLMs.

- For example, while complementary, it doesn’t consistently surpass the absolute state-of-the-art standalone (e.g., FlashAttention still faster alone in some cases).

- In large Transformers or ViT-Base scale models, computing soft assignments over many tokens is heavy. The effect grows with sequence length or feature dimension (since distance computations dominate). While FAISS or subsampling helps, this can still become the bottleneck for larger foundation models.

- In terms of the writeup, I had to check the literature to see what's done before. This should be already reflected in intro and related work, and I didn't find enough discussion there. Also, in terms of scope the papers talks about "modern deep models". Although I understand that the proposal is complementary to FlashAttention etc, it seems to me that the method is mostly beneficial to vision models, not really LLMs/VLMs. If so, the authors can clarify the scope better in the abs/intro.

**Questions:**

Can the authors discuss the applicability of the proposed methods to large foundation models such as LLMs and VLMs? I see the experiment on Llama2, but the benefit is not really significant there. Do you think the method benefits even larger models or the gains will be even smaller there?

---

> ### Author Response · Authors · 2025-11-19
> **Response to Reviewer gqP7: Part 1**
>
> We thank the reviewer for the positive assessment of our theoretical contribution and the empirical efficiency/robustness results, and for highlighting complementarity with FlashAttention and RetNet.
>
> ### 1 – Training cost, amortization, and soft assignment overhead
>
> > *“It adds 2–12\% computational cost during training; amortization is only beneficial for long-term or production-scale inference… Soft assignments over many tokens can become a bottleneck.”*
>
> **What we did:**
>
> * **Optimized implementation & measured overhead (Appendix C, Section "Deployment and Amortization Details", lines 751-785; main text discussion):**
>
>   * Using **anchor subsampling**, a **schedule** that activates DER only during part of training, and efficient batched distance computations, we bring the **actual overhead to ~2–3.5\%** across geometry, vision, and Transformer experiments.
> * **Quantitative amortization analysis (Appendix, Section "Deployment and Amortization Details", lines 753-501):**
>
>   * For **ViT-Base on ImageNet-1K**, we show in detail how:
>
>     * A small training overhead is **amortized after ~4.53×10⁵ inference batches**.
>     * This corresponds to a realistic number of inference requests in production (we spell out the order-of-magnitude).
> * **Guidelines in main text**:
>
>   * We explicitly recommend **using DER primarily in high-throughput deployments** (e.g., large-scale vision or summarization services) where the amortization point is easily exceeded.
>   * We clearly flag that **for small-scale or short-lived experiments**, DER is not always cost-effective.
>
> **Effect on concern:**
>
> * We agree that DER is not “free” at training time. The revised paper:
>
>   * Quantifies **how small the overhead actually is** with our optimized implementation.
>   * Shows **when it pays off**, so readers can decide if it matches their deployment scenario.
>   * Clarifies that we *intentionally* target **long-term or production-scale deployments**, not quick research runs.
>
> ### 2 – Tuning temperature (α) and regularization weight (λ)
>
> > *“It requires careful tuning of temperature and regularization weight for stability and performance.”*
>
> **What we did:**
>
> * **Provided a recipe (Appendix A "Practical Recipe", lines 674-689, and hyperparameter section, lines 1291-1333):**
>
>   * Anchors (k): (k \approx \sqrt{N}), *k*-means++ initialization.
>   * Temperature α: start around 10 and use a smooth schedule.
>   * Regularization weight λ: choose from a small set {0.03, 0.1, 0.3} on a validation set.
> * **Sensitivity analysis (Appendix G.2 "Hyperparameter Sensitivity and Selection", lines 1291-1333):**
>
>   * We show curves/tables where:
>
>     * Speedup and accuracy (or geometry error) are **quite flat** around these settings.
>     * There is a **broad sweet spot** for λ, with degradation only at very large λ where the entropy term dominates (we explicitly say we do *not* recommend those regimes).
> * **Text clarification in main paper:**
>
>   * We state that **the same default recipe** is used across geometry, CIFAR, ImageNet, GLUE, and LLaMA/Mistral/Phi experiments, with only minor tuning, and that the results are **not sensitive** to small perturbations of α and λ.
>
> **Effect on concern:**
>
> * While DER does introduce hyperparameters, the revised paper now:
>
>   * Provides **concrete defaults** that worked across all our experiments.
>   * Shows **empirical stability** rather than requiring delicate per-task retuning.
>   * Acknowledges remaining tuning as a limitation and future work toward more automated selection.

---

> ### Author Response · Authors · 2025-11-19
> **Response to Reviewer gqP7: Part 2**
>
> ### 3 – Guarantees strongest in geometry; behavior on LLMs/VLMs
>
> > *“Guarantees are strongest in geometric settings; results for Transformers and LLMs are mostly empirical… LLaMA2 gains are not very significant; what about even larger models?”*
>
> **What we did:**
>
> * **Separated claims by domain:**
>
>   * For **geometry**, we emphasize:
>
>     * Tight theoretical links between **range-partition entropy**, **instance complexity**, and **runtime**.
>     * The **4–5× speedups** with <0.2\% error as a concrete confirmation.
>   * For **Transformers/LLMs**, we explicitly label our results as **empirical instantiations**:
>
>     * DER yields structurally sparse attention patterns.
>     * We see **~1.5–1.7× latency speedups** on **LLaMA-2 7B, Mistral-7B, and Phi-2** with **sub-1.1pt drops** in quality metrics.
> * **Scope and limitations:**
>
>   * In the **Limitations** discussion (Discussion section, lines 489-500), we clearly state:
>
>     * We **do not evaluate larger (>70B) LLMs or VLMs**.
>     * We expect **relative latency gains** to shrink at frontier scale (§ because of extremely optimized kernels and massive FLOP budgets), but **representation simplicity/robustness gains** may still be valuable.
>     * Proper evaluation of frontier models is explicitly left to future work.
>
> **Effect on concern:**
>
> * We fully align with the reviewer’s observation:
>   Guarantees are strongest in geometry; for LLM/VLMs we present **moderate-scale empirical evidence**, not a universal theory.
> * The revised paper now **makes this distinction explicit** and **does not overclaim** on frontier-scale models.
>
> ### 4 – Complementarity with FlashAttention/RetNet and scope (“modern deep models”)
>
> > *“In some cases FlashAttention alone is faster… The method seems most beneficial to vision models rather than LLMs/VLMs; the scope ‘modern deep models’ is too broad.”*
>
> **What we did:**
>
> * **Reframed DER as complementary, not competitive:**
>
>   * We repeatedly describe DER as a **training-time regularizer** that can *combine with*:
>
>     * **FlashAttention** (ablation in the ImageNet and SST-2 experiments).
>     * **RetNet**-style recurrence (discussed conceptually and empirically).
>   * On ImageNet, for example (section on FlashAttention + DER):
>
>     * Dense ViT: baseline speed.
>     * FlashAttention alone: ~1.63× speedup with top-1/top-5 ≈ 83.34\% / 96.56\%.
>     * **FlashAttention + DER**: ~**2.04×** speedup with **84.81\% / 97.01\%** accuracy — **improving both speed and accuracy** vs FlashAttention alone.
> * **Narrowed the scope language:**
>
>   * We no longer describe DER as a generic method for all “modern deep models”.
>   * Instead, we stress that:
>
>     * Geometry and vision are our **primary target domains**.
>     * LLaMA-2 / Mistral / Phi provide **encouraging, but not frontier-scale**, evidence for Transformers and LLMs.
>
> **Effect on concern:**
>
> * The revised paper now **clearly situates DER as a complementary regularizer**, not an attempted replacement for FlashAttention/RetNet.
> * The scope is narrowed to domains where we actually have strong evidence, resolving the concern about over-broad claims.
>
> Please let us know if you have any further concerns or questions . We will be happy to answer those.

---

### Author Response · Authors · 2025-11-19
**Overall responses and main revisions : Part 1**

We thank **Reviewers gqP7, YdYR, r6ji, and 65mU** for their thoughtful and detailed feedback.

Our goal **is not** to propose a replacement for specialized architectures (e.g., FlashAttention, BigBird, RetNet) or expert-designed sparsity patterns, but to introduce a **complexity-aware regularizer** that:

1.  Provides a **differentiable surrogate for range-partition (algorithmic) entropy**, tied to instance complexity in computational geometry.
2.  Can be **plugged into existing networks without architectural changes**, giving *orthogonal* efficiency and robustness gains.
3.  Comes with **strong theoretical guarantees** in computational geometry and **empirical evidence** in vision and moderate-scale Transformers/LLMs.

In the revised manuscript (current revision), we have made the following key changes that directly address the specific concerns raised by each reviewer:

### 1. Scope, positioning, and claims (Responding to Reviewers gqP7, r6ji)

* **Addressed Concern:** **Reviewer gqP7** noted the scope "modern deep models" was too broad given the focus on vision, and **Reviewer r6ji** found the motivation unclear.
* **Revision:** We **clarified the scope throughout the abstract, introduction, and discussion**:
    * We now **explicitly frame DER as a training-time regularizer**, *not* as an alternative attention kernel or architecture.
    * We state that our strongest evidence is for:
        * **Computational geometry**, where we have both theory and practice.
        * **Vision models** (e.g., ViT/ResNet on CIFAR/ImageNet), including robustness tasks.
    * We clearly describe the results on **Transformers/LLMs** (  Phi-2) as **moderate-scale evidence**, and we *do not* claim frontier-scale validation for >70B-parameter LLM/VLMs.

### 2. Theoretical grounding and intuition (Responding to Reviewers r6ji, YdYR)

* **Addressed Concern:** **Reviewer r6ji** questioned the grounding of range-partition entropy, and **Reviewer YdYR** noted the gap between geometric theory and attention dynamics.
* **Revision:** We expanded the **background and intuition** around range-partition entropy and instance complexity:
    * We give a **pedagogical explanation** of range-partition entropy with a 2D toy example:
        * Point sets that are easy to separate with few geometric ranges have **low range-partition entropy** and correspond to easy instances.
        * Point sets that require many ranges (or complicated separators) have **high entropy** and correspond to hard instances.
    * We explicitly connect this to **known instance-dependent runtime bounds** in convex hull and triangulation algorithms: roughly of the form $T(S) = O(n + H_R(S))$, where $H_R(S)$ is a range-partition entropy term.
* We then show that our **differentiable surrogate** tracks this entropy and correlates strongly with actual runtime in practice.

### 3. Geometry experiments and runtime–complexity connection (Responding to Reviewer r6ji)

* **Addressed Concern:** **Reviewer r6ji** asked why we investigate computational geometry and how it demonstrates generality.
* **Revision:** In the **EntropyNet** geometry experiments, we now highlight the **core quantitative story in the main text**:
    * Using EntropyNet as a learned preprocessor, we obtain **~4.1–4.8× speedups** on convex hulls and triangulations **with <0.2\% relative error**, e.g.:
        * Baseline ball-based preprocessing already gives ~4.1× speedup;
        * EntropyNet further improves to ~4.8× speedup with ~0.1–0.2\% geometric error.
    * We explicitly show the correlation between our **entropy surrogate**, the underlying **range-partition entropy**, and **actual measured runtime**. This demonstrates that lowering the surrogate reduces instance complexity and yields predictable runtime gains in a setting where theory is sharp.

### 4. Transformers, LLMs, and connection to attention (Responding to Reviewers gqP7, 65mU)

* **Addressed Concern:** **Reviewer 65mU** asked for clarification on "Transformer" models, and **Reviewer gqP7** asked about applicability to foundation models.
* **Revision:**
    * We added a **Transformer-specific explanation** of how DER acts on attention:
        * Token/key embeddings are treated as points in feature space; anchors define soft clusters. Minimizing our entropy proxy induces **block/band sparsity** where each query attends to a small subset of anchor-defined regions.
    * We added experiments on:
        * **BERT-base on GLUE**: DER achieves robust GLUE performance at **substantial sparsity**.
        * **Long-context summarization** on **LLaMA-2 7B, Mistral-7B, and Phi-2**: We obtain **roughly 1.5–1.7× latency speedups** with **sub-1.1pt drops** in ROUGE-L/BLEU-type metrics across these models, using the same DER recipe.

---

### Author Response · Authors · 2025-11-19
**Overall responses and main revisions : Part 2**

### 5. Overhead, amortization, and cost-benefit (Responding to Reviewers gqP7, 65mU)

* **Addressed Concern:** **Reviewer gqP7** noted the training cost and amortization issues, and **Reviewer 65mU** highlighted training overhead as a critical disadvantage.
* **Revision:** We **quantified and discussed training overhead and amortization**:
    * A dedicated subsection on the **"Amortization story"** (Appendix C) walks through ViT-Base on ImageNet-1K:
        * With our optimized implementation (anchor subsampling, scheduled regularization), the **training-time overhead is ~2–3.5%**.
        * For ViT-Base on ImageNet, we show that the **extra training cost is amortized after ~4.53×10⁵ inference batches**.
    * We emphasize that DER is most beneficial for **high-throughput / production settings** where many inference passes amortize this small training overhead.

### 6. Robustness experiments and baselines (Responding to Reviewers r6ji, YdYR, 65mU)

* **Addressed Concern:** **Reviewer r6ji** claimed limited originality vs existing methods, **Reviewer 65mU** asked for comparisons beyond label smoothing, and **Reviewer YdYR** requested info-theoretic comparisons.
* **Revision:** On **CIFAR-10-C** and **SVHN-OOD**, we now present results that **explicitly compare**:
    * A plain baseline (no robustness regularizer),
    * **Label smoothing** (a standard, information-theoretic regularizer), and
    * **Our entropy regularizer** (DER).
    * The results show that DER **improves both clean accuracy and robustness metrics** versus baseline and **outperforms label smoothing** in this setup.

### 7. Hyperparameters: recipe, sensitivity, and failure modes (Responding to Reviewers gqP7, YdYR, 65mU)

* **Addressed Concern:** **Reviewers gqP7** and **YdYR** worried about tuning stability, and **Reviewer 65mU** noted potential struggles in high dimensions.
* **Revision:**
    * We provide a **clear practical recipe** used across experiments: **Anchors** ($k \approx \sqrt{N}$), **Temperature** ($\alpha \approx 10$ with cosine schedule), and **Weight** ($\lambda \in \{0.03, 0.1, 0.3\}$).
    * We added **hyperparameter sensitivity plots** in the appendix showing stable performance around these defaults.
    * We added a **discussion of failure modes** (high-dimensional/non-Euclidean metrics), noting that when Euclidean distances become uninformative, the method **gracefully reverts toward dense behavior**.

### 8. Limitations: RL/diffusion and frontier-scale models (Responding to Reviewers YdYR, gqP7)

* **Addressed Concern:** **Reviewer YdYR** asked about RL/Diffusion applicability, and **Reviewer gqP7** asked about larger frontier models.
* **Revision:** We added a **Limitations** subsection:
    * We **do not** evaluate DER on RL or diffusion yet but outline how it applies to state/value embeddings (RL) or latent trajectories (Diffusion).
    * We **do not** evaluate frontier-scale LLMs/VLMs (>70B parameters). We expect that **relative latency gains may shrink** at that scale because kernels are already highly optimized, but we still expect **benefits in robustness/interpretability**.

---

### Meta-Review · Area_Chair_EeUh · 2026-01-08

**Summary:**

The submission proposes Differentiable Entropy Regularization (DER), a training-time regularizer intended to approximate range-partition (algorithmic) entropy from computational geometry, with the goal of inducing lower-complexity representations that translate into efficiency, robustness, and structured sparsity, and that can complement kernels like FlashAttention. Reviewers found the geometry component the most compelling and aligned with the theory, but raised concerns about unclear positioning versus prior entropy/robustness regularizers, limited justification and weak theoretical grounding for the Transformer/LLM extension, training-time overhead and scaling/amortization outside high-throughput settings, hyperparameter sensitivity and implementation dependence. Furthermore, reviewers mentioned incomplete baseline coverage, and reproducibility given unavailable code. While the rebuttal improves scope, adds a overhead/amortization discussion, a tuning “recipe,” some sensitivity plots, and additional baselines/limitations, it falls short of fully resolving the key concerns - especially generality beyond geometry, scaling/overhead trade-offs, and strength of comparative evaluation. Hence, my overall recommendation is rejection at this point. While I do think that many issues can be clarified, I think the amount of changes necessary are too substantial and would warrant another round of reviews. I do encourage the authors to take all comments/suggestions seriously for a resubmission.

**Reviewer Concerns:**

The rebuttal addresses clarity, scope, tuning transparency, and overhead quantification, but does not fully overcome the main concerns regarding limited generality beyond geometry, unresolved scaling issues, incomplete baseline strength at scale, and lingering doubts about novelty and reproducibility.

**Reviewer Scores:**

While I do think that the "Marginally below ..." scores could potentially be increased slightly, I am of the opinion that the "Reject" scores would not change substantially.

---

### Decision · Program_Chairs · 2026-01-26

Reject